

# An Overview on Isotopic Divergences – Causes for instability of Tree-Ring Isotopes and Climate Correlations

Martine M. Savard[1], Valérie Daux[2]

[1]Geological Survey of Canada, Natural Resources Canada, Québec, G1K 9A9, Canada
[2]Laboratoire des Sciences du Climat et de l'Environnement (LSCE/UVSQ/CNRS/CEA/IPSL), Gif-sur-Yvette, 94800, France

*Correspondence to*: Martine M. Savard (martinem.savard@canada.ca)

**Abstract.** Climatic reconstructions based on tree-ring isotopic series convey substantial information about past conditions prevailing in forested regions of the globe. However, in some cases, the relations between isotopic and climatic records appear unstable over time, generating the 'isotopic divergences'. Former reviews have thoroughly discussed the divergence concept for tree-ring physical properties, but not for isotopes. Here we want to take stock of the isotopic divergence problem, express concerns and stimulate collaborative work for improving paleoclimatic reconstructions.

There are five main causes for divergent parts in isotopic and climatic series. (1) Artefacts due to sampling and data treatment, relevant for dealing with long-series using sub-fossil stems. (2) Stand dynamics, including juvenile effects mostly occurring in the early part of tree-ring series. (3) Rise in atmospheric $pCO_2$, which can directly influence the foliar behaviour. (4) Change of climate, which may modify the isotope-climate causal links. Finally (5), atmospheric pollution, which may alter leaf and root functions. Future paleoclimate research would benefit from interdisciplinary efforts designed to develop further process-based models integrating multi-proxy inputs, so to help identify causes of isotopic divergences and circumvent some of them in inverse applications.

**Keywords**: carbon isotopes, oxygen isotopes; climate reconstruction; response functions; transfer functions

## 1 Introduction – Divergence, informative but knotty

Tree-ring isotopes can serve as proxies of climatic parameters for reconstructing past climate variability, which is useful for understanding regional and global climatic patterns (Treydte et al., 2007; Braconnot et al., 2012; Naulier et al., 2015a). In most cases, such reconstructions assume that the observed modern statistical relationship between tree-ring isotopic proxies and measured climatic parameters was identical in the past. Parameters such as ambient temperature, solar radiation and air moisture are also assumed to trigger responses in tree biological functions (e.g., stomatal conductance), thus modulating quantifiable tree-ring stable isotopes of carbon ($\delta^{13}C$) and oxygen ($\delta^{18}O$). However, tree physiological reactions to changes in environmental conditions do not behave linearly (e.g., Schleser et al., 1999), and several influencing factors may interplay and



alternate during the lifetime of a tree. When correlations between climatic parameters and tree-ring proxies show periods of instability such that correlations weaken, become non-significant or change in signs, the relationship between proxies and climatic data shows a 'divergence'. The concept of the divergence problem first introduced for the offset between ring-width and instrumental temperatures of recent decades appears in a large body of literature (e.g., Jacoby and D'Arrigo, 1995; Briffa et al., 1998; D'Arrigo et al., 2008; Esper and Frank, 2009). We refer hereafter to this weakening of the response of tree-ring

growth or density to temperature or other climatic parameters as 'growth divergence'.

Physical characteristics (width, density) and isotopic attributes ($\delta^{13}$C, $\delta^{18}$O) of tree rings show distinct sensitivities to climatic and non-climatic conditions (Gagen et al., 2006; Brugnoli et al., 2010; Daux et al., 2011; 2015a; Savard et al., 2020). When addressing growth-climate relationships, potential causes of growth divergence include moisture stress, complex non-linear or

threshold responses, changes in season duration, phenology and local pollution (D'Arrigo et al., 2008 and references therein). In addition, detrending ring growth series appears as a methodological potential cause for growth divergence (Esper and Frank, 2009; Esper et al., 2010a). Are those same factors also causing divergence between isotopes and climatic parameters?

The present article deals with the 'isotopic divergence', which we define here as the middle- to long-term (>10 years) loss or

change in signs of correlations between a climatic parameter and tree-ring isotopic ratios ($\delta^{13}$C, $\delta^{18}$O, or rarely $\delta^{2}$H). In this definition, we exclude short-term (high frequency) deviations. Such deviations could derive from: insect outbreaks (up to 7 years; Mayfield III et al., 2005; Simard et al., 2008; Gori et al., 2014), fungi epidemics (3-4 years, Saffell et al., 2014; Lee et al., 2017), forest clearing, pruning, wind throws and felling (2-7 years of increasing discrimination after thinning; Di Matteo et al., 2010). These brief deviations merely juxtapose on long-term changes of climate variability (low frequency, multi-decadal

to multi-centennial). In practice, averaging isotope ratios from several trees and sites smooths out these short-term deviations, which contributes to the uncertainty of the inferred long-term responses.

Isotope-climate divergences *per se* are rarely discussed in the literature (Aykroyd et al., 2001; Naulier et al., 2015b; Daux et al., 2011). However, several reported cases of non-stationary relationships between climatic parameters and the isotopic

proxies altering the skill of climate reconstruction models undoubtedly fall under this definition (Table 1). At present, no review on tree-ring isotopes synthesises the extent and main features of isotopic divergences, although their potential influences on statistical modeling and ensuing climatic reconstructions are clear. One should not ignore that determining periods of isotopic divergence provides key information for understanding climatic patterns and changes in climatic regimes. At the same time, producing unflawed climatic reconstruction requires getting around isotopic divergences.


Given the need for careful assessments of isotopes as climate proxies for various regional contexts and tree species, this synthesis of the up-to-date information on isotopic divergences aims at: (1) describing the main isotopic divergence types and discussing their potential causes, and (2) reviewing research avenues to identify them and account for them (Table 2).

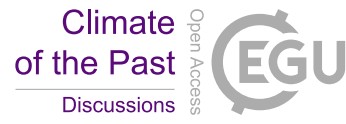

## 2 Mechanisms of tree response to environmental conditions

### 2.1 Control mechanisms on tree-ring isotopes

The well accepted equations of Farquhar et al. (1989) describe the main controls on leaves carbon (C) isotopic values ($\delta^{13}C_L$) and their utility in determining the intrinsic water use efficiency (iWUE, the amount of C acquired per unit of water lost), which equals the ratio of the photosynthetic rate (A) to the gaseous conductance (g) as follows:

$$\delta^{13}C_L = \delta^{13}C_{at} - (a + (b-a) \times c_i/c_{at}); \qquad iWUE = A/g = c_{at} \times (b - \delta^{13}C_{at} - \delta^{13}C_L)/ 1.6 \times (b-a) \qquad \text{(Eqs. 1, 2)},$$

where $\delta^{13}C_{at}$ is the value of atmospheric $CO_2$, 'a' is the fractionation due to gaseous diffusion of $CO_2$ entering the foliar system via stomata, 'b' is the enzymatic fractionation during assimilation of $CO_2$ (carboxylation), and $c_i$ and $c_a$ are the intra foliar and atmospheric pressures of $CO_2$, respectively. Several post-photosynthetic processes can also modify the final $\delta^{13}C$ values fixed in tree rings through C exchange with stored carbohydrates (Gessler et al., 2014). However, tree-ring $\delta^{13}C$ values ($\delta^{13}C_{TR}$) generally vary in parallel with $\delta^{13}C_L$, but with an offset (Loader et al., 2003; Verheyden et al., 2005).


The $\delta^{13}C$ change of atmospheric $CO_2$ due to the addition of $^{13}C$-depleted $CO_2$ from fossil fuel combustion since the beginning of the industrial era (Suess effect) is readily overcome by removing the related isotopic deviation from the $\delta^{13}C$ series (McCarroll et al., 2009). In this article, we mostly discuss $\delta^{13}C$ values already corrected for the Suess effect, except when providing information on the ecophysiological models that use raw $\delta^{13}C$ values as input.


Several equations describe the $^{18}O$ enrichment in leaves due to complex fractionation steps during oxygen (O) assimilation by trees. But a simplified version of these equations (Barbour, 2007) expresses the main controls on the final leaf water ratios ($\delta^{18}O_f$), which include the soil water signal ($\delta^{18}O_S$), relative humidity (RH), the proportion of source water from the roots in leaves ($f$), equilibrium fractionation factor ($\varepsilon^+$), kinetic fractionation factor ($\varepsilon_k$) and average water and carbonyl group exchange

fractionation ($\varepsilon_{wc}$):

$$\delta^{18}O_L = \delta^{18}O_S + (1-f) \times (1-RH) \times (\varepsilon^+ + \varepsilon_k) + \varepsilon_{wc} \qquad \text{(Eq. 3)}.$$

When dealing with tree-ring $\delta^{18}O$ cellulose, the dampening factor reflects the proportion of O exchanged with source water when sucrose is broken down into hexose phosphates during cellulose synthesis, allegedly constant in time (Ogée et al., 2009; Cernusak et al., 2016). Hence, processes described in the Craig-Gordon model will influence the final tree-ring cellulose values

through the stomatal functions (Roden et al., 2000; Cernusak et al., 2016; Belmecheri et al., 2018).

Any direct or indirect climatic factors operating on the photosynthetic or respiratory functions of trees modify the A, g, A/g ratio or O fractionation factors (Eqs. 1, 2), and consequently act upon the tree-ring $\delta^{13}C$ and $\delta^{18}O$ values. The sensitivity of these proxies to changes in temperature, precipitation, RH and light (cloudiness, radiation) vary with the species of trees, site

conditions and regional climate.



## 2.2 Linking isotopic time series to climatic parameters

Amongst the approaches linking the isotopic responses to meteorological variations (calibration), several statistical methods exist, with various degrees of complexity, of which most assume that the relationships are linear. The strongest, significant correlations between an isotopic series representative of a tree population and monthly or daily meteorological data sets

adequate for the region of interest help target the climatic series to reconstruct. Multiple linear regressions constitute the most widespread statistical approach that serve fitting the relationship (response function) between the tree-ring isotopic series and the selected regional climatic data (Fig. 1). To assess the robustness of the calibration model, cross-validation techniques use sub-sets of the isotopic series corresponding to the meteorological instrumented period. One set serves calibrating the climate signal (root mean squared error - RMSE; coefficient of determination, $R^2$), and the other set, for validating its reconstruction

skills (reduction of error, RE; coefficient of efficiency, CE; Briffa et al., 1988). The validated transfer function helps reconstructing back the climatic parameter over periods when the climatic information is not available (Fig. 1).

Another methodological approach builds on the Bayesian principles for reconstructing one climatic parameter using multiple proxies over a calibration period when the set of proxies and the measured climatic parameter overlap. A model developed for

each proxy estimates the likelihood of the proxy to have a specific value for a given datum of the climatic parameter (details in Tingley and Huybers, 2009; Kruschke, 2010). The prior and posterior distributions of the Bayes theorem allow predicting the climatic parameter from each proxy. This approach allows reducing the impact of individual proxy errors, accounting for uncertainties, and running sensitivity tests for assessing the different proxy responses to a specific forcing (Tolwinski-Ward et al., 2013; Emile-Geay and Tingley, 2015). Combining tree-ring $\delta^{18}O$, width and density series of spruce trees from northern

Quebec, a Bayesian approach produced an improved millennial temperature reconstruction compared to the ones obtained from individual proxies (Gennaretti et al., 2017).

Ecophysiological models also called vegetation, biophysical, process-based, mechanistic or tree-growth process models relate numerous mechanisms to multiple measured foliar or tree-ring proxies (Eqs. 1-3). These models allow reproducing tree growth

on a daily basis by integrating tree-ring width and cellulose isotopes with daily environmental data such as minimal and maximal air temperatures, amount of precipitation, atmospheric $CO_2$ concentration and $\delta^{13}C$ values (e.g., Ogée et al., 2009; Danis et al., 2012). The different mechanistic models have various structures, use different assumptions and calibration methods, and have differing sensitivity to climatic or non-climatic triggers (e.g., Guiot et al., 2014). Most models make forward predictions and allow verifying that the measured tree-ring isotopic trends compare well with the isotopic outputs modelled

with the meteorological and non-meteorological inputs. So far, only MAIDENiso has been used in inverse mode for reconstructing climate (Danis et al., 2012; Boucher et al., 2014; Lavergne et al., 2017). The limitations when using process-based models come from the fact that all the required daily input data are in cases not measured over long periods, or in other cases, just derived, inducing uncertainties. Yet, two advantages of process-based modelling for climate reconstruction are that



they allow running a wide range of sensitivity analyses for highlighting the most influential mechanisms, and they integrate

the atmospheric $CO_2$ concentration and $\delta^{13}C$ values as inputs (see also Section 3.3). Also considering process-based approaches, climatologists refer to the so-called proxy-system models (e.g., the Vaganov-Sashkin-Lite - VS-Lite - growth model; Sánchez-Salguero et al., 2017). A proxy system links environment (environmental triggers) to observations as depicted by sensors (proxy responses) through models of low, intermediate or high degrees of complexity depending on the objectives and need of a specific research program (Evans et al., 2013). The tree proxy system can use tree-ring (archive) isotopes (proxy)

with or without integrating other tree-ring proxies and reconstruct, interpret climate and determine the uncertainty of the results. Researchers can combine the tree proxy system with several other proxy systems such as pollens, lake sediments, corals, and ice cores investigated at local scale to perform climate modeling at broad scale (Dee et al., 2016; Okazaki and Yoshimura, 2019).

Understanding past global climatic patterns using isotopes in trees requires producing tree-ring series from living trees and sub-fossil stems covering several centuries or millennia (e.g., Treydte et al., 2006; Gagen et al., 2011; Labuhn et al., 2014; Naulier et al., 2015b; Helama et al., 2018; Giguère-Croteau et al., 2019; Kłusek et al., 2019; Wang et al., 2016). During these long periods, climatic forcings are also in motion; they will possibly modify the dominant factors controlling isotopic responses in trees. Hence, non-linearity of relations between tree-ring isotopes and climatic parameters due to biological functions or

abiotic external changes may weaken the statistical links and create isotopic divergences. In order to produce reconstructed time series of past climate variability devoid of parts not reflecting climate reality, any approach selected for modelling the climate isotopes-based reconstruction must consider isotopic divergences.

## 3 Preventable types of climate-isotope divergence

### 3.1 Departures due to sampling and data treatments

Potential artefacts in isotopic series may arise due to sampling procedures and data treatment when building multi-centuries or millennial series from sub-fossil stems or timber assemblages. In this case, the selection of stems pertains to their wood quality and availability in sufficient numbers. Joining two successive cohorts of randomly picked stems may be problematic due to possible isotopic offsets between cohort averages. Shifts between cohorts if not adjusted, or adjusted improperly, can generate long-term trends in the isotopic record and yield a biased climatic reconstruction.


An approach to overcome the problem of isotopic departure between cohorts when the replication is low consists in evaluating if there are offsets between averages of successive cohorts designed to overlap over 5-years (or more if judged necessary). The next step involves calculating the isotopic means for a minimum of 10 additional individual stem segments covering the existing overlap between two successive cohorts (Gagen et al., 2012; Naulier et al., 2015a). This procedure repeated for every

cohort junction was applied for joining shifted 5-year blocks of sequential pooled series, but it could apply to junctions of any



cohort types, such as averaged individual series, pooled yearly-sampled series or combined types (e.g., Foroozan et al., 2019). The initial junction point is set from the overlap of the youngest stem cohort with a living-tree cohort. If an offset exists, the development of a linear regression permits estimating a factor for correcting a cohort between its two points of junction, adjusting the series from the most recent cohort to the oldest. In spruce trees from northeastern Canada, only two junction
points out of 12 needed correction (Naulier et al., 2015a). The applicability of this correction-factor method depends on the availability of a large number of stem segments, a particular challenge when dealing with fossil material laboriously collected from lakes, or with construction beams from monuments where the coring allowed is strictly limited.

If the number of stem segments available for research is limited, adjusting cohorts to one another, without additional individual
series for calculating a factor of correction is one potential approach (Hangartner et al., 2012). One variation of this approach implies standardizing each cohort to a mean of zero and a standard deviation of one (z-score). Another variation simply adjusts the average value of the older cohort to the overlap average of the newer cohort (Kress et al., 2014; Labuhn et al., 2016). The standardization method involves loosing low-frequency variability that exceeds the length of the cohorts, whereas the average-adjustment keeps the low frequency records but could produce artificial long-term trends in the isotopic series (Hangartner et
al., 2012). Still in the case of low availability of stem segments, another approach is to combine ring isotopic series from different sites, but this approach may induce biases that need correction through methods such as regional curve standardization (RCS; Helama et al., 2018) or other detrending methods.

Often the number of trees considered for paleoclimate reconstruction is five (e.g., Leavitt, 2008). However, one should keep
in mind that the best approach for avoiding divergences due to isotopic artefacts when investigating low frequencies relevant for long-term climate reconstruction is ideally to combine several stem cores to limit artefacts due to intra-ring variability (e.g., Esper et al., 2020). In addition, studies of living trees indicate that the mean for 10 or more individual series yields adequate performances in terms of $\delta^{13}C$ and $\delta^{18}O$ ranges and confidence levels, and preserves reliable low frequency climatic records (Loader et al., 2013b; Daux et al., 2018). If low replication only is possible, adjustments of the composite series are required.
In the end, adjusting the cohort values can overcome large divergences potentially created by using several cohorts of low replication in constructing multi-centennial series, but the possibility of false isotopic correlations with climatic parameters still persists after joining the cohorts into composite series.

### 3.2 Height and stand dynamics

The ambient conditions and the response of young and short trees may differ from those of more mature and, most importantly,
taller trees. Generally, trees progressively evolve from the understorey to the canopy during their growth, which makes them gradually experience warmer temperature, lower humidity (higher vapour pressure deficit, VPD), better access to solar radiation, and higher exposition to wind (Freiberg, 1997; Zweifel et al., 2002). This development may increase cellulose $\delta^{13}C$ and $\delta^{18}O$ values by enhancing the photosynthesis and transpiration rates with tree age (Banerjee and Linn, 2018). While





distance between their foliar system and the soil increases, trees leave an atmosphere containing respired $^{13}$C-depleted $CO_2$.
They also progressively access the open atmosphere, where $CO_2$ is $^{13}$C-rich, generating a positive trend in the cellulose $\delta^{13}$C with time (Francey and Farquhar, 1982; Schleser and Jayasekera, 1985). In open canopies, the ambient conditions as trees get taller are relatively stable, and the isotopic effects described above are very likely limited to non-existent (Brienen et al., 2017; Klesse et al., 2018). However, the height increase imposes hydraulic limitation and possibly reduction of stomatal conductance, which may lead to a rise of the cellulose $\delta^{13}$C values with age (Brienen et al., 2017, and reference herein). In certain cases, the
mean depth of tree rooting may increase with the size and age of specimens, at least over the first years of growth (Weltzin and McPherson, 1997; Bouillet et al., 2002; Irvine et al., 2002; Ma et al., 2013). The absorption of an increasing proportion of deeper, less evaporated, and therefore more $^{18}$O-depleted source of water may result in a negative trend in cellulose $\delta^{18}$O series (Dawson, 1996). These developmental changes, and their possible impacts on $\delta^{13}$C and $\delta^{18}$O ratios, take place during the early life of the trees, when trees grow up at a maximum rate. However, the duration of this 'juvenile' effect is highly variable. For
instance, small and short-lived (<0.5‰ over 5 years; Duffy et al., 2017), to moderately intense and long $\delta^{18}$O effects (1.2 ‰ increase over 10 years; Labuhn et al., 2014) are reported for oak cellulose. But a majority of studies conclude to an absence of juvenile effects on $\delta^{18}$O series (e.g., Raffalli-Delerce et al., 2004; Porter et al., 2009; Daux et al., 2011 ; Young et al., 2011; Li et al., 2015; Kilroy et al., 2016). To the contrary, juvenile effects are the norm for $\delta^{13}$C values. Many studies dealing with a variety of species and sites reported long increasing trends in the first 20 to 50 years of cellulose $\delta^{13}$C series (Bert et al., 1997;
Duquesnay et al., 1998; Arneth et al., 2002; Li et al., 2005; Gagen, 2008; Labuhn et al., 2014). But no effect was detected in larches growing in open canopies (Daux et al., 2011; Kilroy et al., 2016). To avoid integrating possibly flawed portions in isotope-based climate reconstructions, the first 20-50 years of the isotopic series are frequently truncated (Gagen et al., 2007; Loader et al., 2013a ; Labuhn et al., 2014).

In most cases, stable isotopes in tree rings do not appear to contain long-term age effects beyond the juvenile phase. Notwithstanding, several century-long $\delta^{18}$O trends were reported. For instance, negative 250-year long trends in *Juniperus turkestanica* from Pakistan (Treydte et al., 2006), and 400-year long ones in *Pinus uncinata* from the Spanish Pyrenees (Esper et al., 2010b) were attributed to the increase of the contribution of $^{18}$O-depleted water from deeper soil layers as trees aged. In contrast, 150-year positive trends in *Picea abies* and *Fagus silvatica* (Central Europe; Klesse et al., 2018) were ascribed to the
combined effects of enhanced hydraulic resistance and increased VPD, as trees got taller and accessed canopy. The distance to the upper canopy, which controls not only humidity but also light availability and therefore the photosynthetic capacity and the $\delta^{13}$C values of leaves, appears as the best predictor of the long trends for the two mentioned species at a given site. The tree-ring $\delta^{13}$C series of sub-fossil *Pinus sylvestris* from Northern Fennoscandia reveal even longer trends, which last throughout the tree lifespan (Helama et al., 2015). However, some of the observed trends likely derive from the method of series
construction (see Section 3.1), averaging isotopic data from lake sub-fossil wood of multiple sites slightly differing in environmental conditions (Helama et al., 2018).



The biases described above may induce divergences between isotopic and climatic records. The methods frequently applied for tree-ring width and density proxies such as the RCS or other detrending procedures (i.e. negative exponential function) may help remove these divergences (Esper et al., 2010b; Helama et al., 2018). A high degree of sample replication and the avoidance of pooling are also recommended to ensure that low frequency trends are adequately understood and characterized (Klesse et al., 2018).

### 3.3 Physiological effects of rising $pCO_2$

Beyond the isotopic Suess effect on tree-ring $\delta^{13}C$ values, industrialization has generated the largely recognized foliar physiological effect solely due to the increasing atmospheric pressure of $CO_2$ during the 20[th] century. Independently of climatic conditions, this pressure effect modifies the gas-exchange functions in leaves. The foliar reaction lowers g, or modifies A and g (Franks et al., 2013), and hence imprints the $\delta^{13}C$ values of trees as underlined by Eqs. 1-2. The general direct effect is to increase the photosynthetic discrimination against $^{13}C$, and, as consequence, diminish the $\delta^{13}C$ values of foliar sugars (e.g. Schubert and Jahren, 2012). The effects seems minimal on the tree-ring $\delta^{18}O$ values (Saurer et al., 2003, Battipaglia et al., 2013). The reactions generating the lowest $\delta^{13}C$ values occur if the $c_i$ level increases proportionally with $pCO_2$ ($c_{at}$-$c_i$ constant, passive response), and the lowest effects on the tree-ring $\delta^{13}C$ values occur if $c_i/c_{at}$ stays constant (active response; Saurer et al., 2004). The phenomenon is especially marked after 1955, above 330 ppm of $CO_2$, when the pressure rise is acute (e.g., Waterhouse et al., 2004). But cases of no isotopic responses to rising $pCO_2$ exist as well ($\delta^{13}C$ constant; Silva and Horwath, 2013; Belmecheri et al., 2014; Wieser et al., 2016; Savard et al., 2020).

Over the last decade, the awareness of this effect on $\delta^{13}C$ series has spread widely and most scientists reconstructing climatic parameters using $\delta^{13}C$ values opt to correct these ratios to minimize biased $\delta^{13}C$-climate correlations (e.g., Andreu-Hayles et al., 2017). Several methods to obtain pre-industrial tree-ring $\delta^{13}C$ series apply proportional corrections to rising $pCO_2$ ranging between 0.0073 and 0.02‰/ppm (Feng and Epstein, 1995; Kürschner, 1996; Saurer et al., 2003, Konter et al., 2014). The performance of the proportional correction may improve by testing the reproduction of instrumental climatic series and adapting a corrective factor specific for the investigated region (Treydte et al., 2009). A widespread corrective approach uses a conditional, pre-industrial (pin) correction throughout six steps (McCarroll et al., 2009). This non linear detrending of the low-frequency changes better works when the measured $\delta^{13}C$ series starts before or at the beginning of the industrial period (1850), otherwise the method might under-correct the $\delta^{13}C$ values (Schubert and Jahren, 2012). As in the proportional correction, the pin model assumes that the tree responses to rising $pCO_2$ are linear and uniform, even if sometimes the responses are non-linear for a given tree and heterogeneous amongst trees (Waterhouse et al., 2004; McCarroll et al., 2009). The approach pointedly applies a non-linear regression to trees, a step that considers the $pCO_2$-induced response specific to each tree.



However, there is no overarching consensus as to which corrective method to apply to bring $\delta^{13}C$ series back to the pre-industrial level (Treydte et al., 2009; Konter et al., 2014). A wise approach is to test the various corrective methods and assess the performance of the resulting series with climatic reconstruction models. In general, using the mentioned corrections removes the effects of rising $pCO_2$ of the last 170 years. Exceptions to this rule can occur if rising $pCO_2$ plays concomitantly with other natural factors such as loss of nutrients. Indeed, a case of possible nutrient loss generated an extreme active reaction

to rising $pCO_2$ and an anomalous $\delta^{13}C$ series at a xeric site (Giguère-Croteau et al., 2019). An interesting recent development for addressing the $pCO_2$-related isotopic divergence is through multi-proxy ecophysiological modeling. A process-based model (MAIDENiso) applied to tree-ring width and isotopic series from oak species exemplifies this type of thorough approach, which optimizes temperature reconstructions by including increasing $pCO_2$ directly into the model (Boucher et al., 2014).

In summary, foliar physiological reactions to rising $pCO_2$ may generate departures in climate-$\delta^{13}C$ correlations globally, but corrective methods applied routinely to calculate pre-industrial $\delta^{13}C$ series largely minimizes this effect, and yet the scientific community still debates upon finding an unanimous corrective approach (e.g., Konter et al., 2014). The remaining major problems for producing reliable statistically reconstructed climatic series originate from two main types of isotopic divergences: climate change (long-term shift in climatic regimes) and pollution stress. Section 4 reviews the causes of these

potential impeding isotopic divergences, and reviews prospective avenues for correcting them (Table 2).

## 4 Critical causes of climate-isotopes divergence and suggested corrective measures

### 4.1 Climate change

#### 4.1.1 Switching climatic controls

Multivariate environmental factors modulate C and O isotopic fractionation in trees and the $\delta^{18}O$ and $\delta^{13}C$ values of tree-ring

cellulose can record these modulations. Generally, statistically significant correlations exist between the predominant factors and the isotopic records. However, the isotopic response to the climate forcing may vary over time because changes in climatic regimes regulate the relative influence of the parameters that interplay in generating tree-ring isotopic signatures. For instance, because of the coupled and counteractive influence of moisture and temperature on the tree-ring $\delta^{13}C$ values in *Abies alba* from the Black forest (Germany), the $\delta^{13}C$ - relative humidity and $\delta^{13}C$ - temperature correlations depend on the temperature-

humidity relationship (Edwards et al., 2000). Therefore, for past periods with moisture-temperature relations differing from the one of the calibration period, reconstructed humidity or temperature estimates can diverge from the real values. Some environments of low-moisture stress provide another good example of the effect of regime change, where tree-ring $\delta^{13}C$ values depend primarily on sunshine, hence $\delta^{13}C$ relations with temperature are stable insofar as sunshine and temperature strongly correlate. Yet, studies in northwestern Norway (Young et al., 2010) and the Northern boreal zone (Seftigen et al., 2011)



depicted divergences between temperature records and $\delta^{13}C$ series of *Pinus sylvestris* during episodes of decoupling between irradiance and temperature linked to changes in large scale atmospheric circulation. As illustrated in these examples, when the assumption of stationarity of the temperature-sunshine relation does not stand, reconstructing sunshine or cloudiness rather than temperature is a reliable alternative.

Modifications in atmospheric circulation, which impart changes in the origin and trajectory of cloud masses, can also induce temporal variations of the $\delta^{18}O$ values of rain echoed in tree-ring cellulose at a given site, and independent of climatic conditions *per se* (Saurer et al., 2012; Sakashita et al., 2018). Therefore, if the $\delta^{18}O$ signal of the source water near trees evolves with time, the tree-ring $\delta^{18}O$ series can diverge partly from the climate records. In this way, the nonstationary nature of the relationship between the tree-ring $\delta^{18}O$ values of Alpine *Quercus petraea* (Switzerland) and climate may be ascribed to

variations in moisture source determined by the dominant atmospheric circulation pattern in Europe, that is the North Atlantic Oscillation (Reynolds-Henne et al., 2007). Similarly, the weakening of the earlywood $\delta^{18}O$ response to climate (temperature, relative humidity - RH, VPD) of *Abies forrestii* from Southwestern China may be attributable to changes in atmospheric circulation patterns linked to Pacific sea surface temperatures (An et al., 2019). Another effect of the change of source water is the modification in the strength of the linkage between the $\delta^{13}C$ and $\delta^{18}O$ ratios. A deterioration of this linkage was observed,

for instance, in tree rings of *Sabina przewalskii* in the Tibetan plateau, and attributed to the variation of the source water isotopic composition due to interactions between East-Asia monsoons and westerly circulation (Wang et al., 2016). Hence, in that context, any correlation of $\delta^{18}O$ series with climatic parameters would vary through time. As a final note, climate change through modifications of the timing and duration of the growing season, owing to phenological adaptation can also modify the correlations between tree-ring $\delta^{18}O$ or $\delta^{13}C$ values and climate parameters as summarised below

**4.1.2. Effects on phenology and physiology**

The effects of climate change on tree physiology are numerous and species and site dependent. Our objective is not to make a review of all these effects but to stress some possible physiological responses to climate change, which may induce divergences between isotopic series and climatic records.

Tissue growth starts with budburst, a key process initiating the photosynthetic period. In mid-latitudes, the timing of budding and other spring phenological events of plants (leafing or flowering) largely depends on the air temperature of previous weeks/months (e.g. Defila and Clot, 2005). An effect of global warming has been to advance these spring phenological events in recent decades, from several days up to about 2 weeks (e.g., Walther et al., 2002; Menzel et al., 2006; Fu et al., 2014). Some ecological studies also report delayed autumnal phenological events (growth cessation, bud set and leaf senescence; Walther

et al., 2002; Menzel et al., 2006).



The first report on the sensitivity of correlations between isotopic and climatic records to phenological changes showed that English oak ring $\delta^{13}$C series and temperature correlated optimally if temperature was averaged over a fixed-length period of 20 days (Aykroyd et al., 2001). Averaging used start dates varying with the second flush of leaves, i.e., using a 20-day period

moving within July and August. This example suggests that the application of transfer functions based on isotopic correlations with climatic data of fixed periods of the year can lead to reconstruct climatic parameters with differing statistical significance over time, depending on the strength of the relationship. In other words, isotope-climate correlations using fixed-date intervals may generate divergences. A few examples include: divergences between summer temperature and $\delta^{13}$C series of *Pinus sylvestris* from Eastern Finland in the second half of the 20th century (Hilasvuori et al., 2009); July-August mean temperature

and $\delta^{13}$C and $\delta^{18}$O values of *Larix decidua* from the French Alps since the 1990s (Daux et al., 2011); and maximum summer temperature and $\delta^{18}$O values of *Picea mariana* in Québec (Canada) since 1995 (Naulier et al., 2015b). The overall lengthening of the growing period may affect the relation between $\delta^{13}$C and climate series also because it modifies the tree-ring $\delta^{13}$C ratio. Nonstationary relationships between temperature and precipitation with the $\delta^{13}$C series of oaks from Switzerland may derive from precocious and/or late C uptakes relative to the regular growing periods of trees (Reynolds-Henne et al., 2007). This

interpretation invokes the seasonality of the atmospheric $CO_2$ $\delta^{13}$C signal (Eq. 1), high during summer and low during winter. During growing seasons longer than the regular ones, trees assimilate larger proportions of light C. Several consecutive years of lengthened growth seasons can thus induce long-term decreasing $\delta^{13}$C trends.

Trees use C through direct (assimilation from atmosphere) and indirect (internal storage) pathways to build carbohydrates

during metabolic processes. At the beginning of spring, deciduous trees utilize stored starch and sugars to form early wood. After budburst, photosynthesis directly produces carbohydrates. The proportion of direct assimilates increases progressively at the expense of reconverted stored material, until they are the only carbohydrate source for building new plant tissues and storing reserves, mainly as starch (Kimak and Leuenberger, 2015). Although evergreen conifers rely less on C reserves than deciduous trees, recent photosynthates supply their growth, but C fixed during previous years can also contribute (von Arx et

al., 2017; Castagneri et al., 2018). Detecting old C compounds in the current year C load by using wood or cellulose [14]C analyses highlights this contribution (Gessler and Treydte, 2016). Carbohydrate reserves are generally enriched in [13]C relative to new photosynthates due to post-photosynthetic processes (Damesin and Lelarge, 2003; Cernusak et al., 2009; Werner and Gessler, 2011). Therefore, the use of stored C for trunk growth leads to higher $\delta^{13}$C in wood. Remobilization of stored C can thus have strong effects on the intra-annual $\delta^{13}$C signal (Offermann et al., 2011). Stress factors, such as drought and heat,

impair photosynthesis and can modify the storage and remobilization patterns. Long-term exposure to drought may therefore trigger reoccurring needs to use stored C (Gessler and Treydte, 2016). This effect may uncouple the tree-ring $\delta^{13}$C signal from actual climate and produce divergence between the two series. Water stress can also promote the stomatal control of isotopic fractionation (Cornic, 2000). That way, when the moist Batang-Litang plateau in western China started to experience recurrent




droughts in the 1960s, the $\delta^{13}C$ response of *Abies georgei* to temperature and precipitation progressively changed due to the
gradual transition to stomatal control over the photosynthetic rate (Liu et al., 2014).

During cellulose synthesis, leaves export and exchange some O of sucrose with non-enriched xylem water (Sternberg et al.,
1986). The fraction of O atoms that exchange ($P_{ex}$) equals 42% on average. However, this proportion may vary over the
growing season or over longer periods (Gessler et al., 2009). For instance, $P_{ex}$ appears to depend on environmental conditions
and phenology in *Larix decidua* (Gessler et al., 2013) and to increase with increasing aridity in eucalypt species (Cheesman
and Cernusak, 2016; Belmecheri et al., 2018). In this last case, post-photosynthetic processes had a dampening effect on wood
cellulose $\delta^{18}O$ values, which induced a discrepancy between cellulose $\delta^{18}O$ measurements and RH, proportional to the latter.
Environmental changes (such as increasing aridity over time) may intensify post-photosynthetic exchanges, leading with time
to the decoupling between cellulose $\delta^{18}O$ series and climate. This little studied cause of divergence requires further
investigations.

We mentioned in Section 4.1.1 that divergence between climate and $\delta^{18}O$ records may arise if the root water-uptake deepens
with tree age, because deep soil layers tend to contain $^{18}O$-depleted water relative to surficial soil affected by evaporation.
Such a deepening of rooting depth may also relate to physiological adaptation if trees need to cope with decreasing precipitation
or increasing temperature (Brunner et al., 2015). Indeed, when soil-moisture declines, which often accompanies higher
temperature and evaporative demand from the atmosphere, trees may have to probe down to humid layers or even to the
saturated zone (Fan et al., 2017). As an example, the ring $\delta^{18}O$ series of *Pinus halepensis* from Greece, under drought
intensification, decreases since the 1970s due to an enhanced contribution of depleted deep water to the tree source water. This
long-term $\delta^{18}O$ decline was divergent with the concomitant rise in temperature (Sarris et al., 2013).


During summer middays, high VPD often reduces stomatal closure and hinders $CO_2$ assimilation in tropical (Ishida et al.,
1999), Mediterranean (Raschke and Resemann, 1986), and even cool temperate regions (Kamakura et al., 2012). As long as
the yearly mean duration of the midday depression is short and varies little over time, cellulose is a trustable archive for
isotope-based climate reconstruction. However, when the decline of the photosynthetic rate is severe (i.e., 60%; Kets et al.,
2010), a reduction of sugar production during this midday depression signifies that all daily conditions may not be imprinted
in the isotopic composition of sugars, and consequently, of cellulose. If this phenomenon persists over several days during the
growing season, cellulose becomes a biased, incomplete, recorder of the diurnal environmental conditions. One can foresee
that the global rise of temperature might exacerbate the photosynthetic midday depression, generate information loss in the
cellulose isotopic composition, and induce isotopic divergence; a phenomenon that the literature does not report yet.



## 4.2 Approaches to avoid isotopic divergences due to change in climatic regimes


Eliminating all possible causes of decoupling between isotopic and climatic records due to climate change is very challenging, but selected approaches may minimize the risks of divergence. For $\delta^{13}C$-based climate reconstructions, dealing with the issue of possible effects due to remobilized C for trunk growth may matter. A widespread strategy consists in separating latewood from earlywood whenever possible and analyze latewood alone to avoid carry-over effects from reserves (e.g., Kagawa et al.,

2006). From another angle, during wood formation, xylem cells formed by the cambium pass through successive differentiation stages, namely cell enlargement, cellulose and hemi-cellulose deposition into secondary cell walls, lignification and cell death (Rathgeber et al., 2016). Seasonal interactions between climate and the phenology of wood formation influence the pace and intensity of these phases (Cuny and Rathgeber, 2016). A pioneer study illustrates clearly this innovative concept, as the relations between climate drivers with $\delta^{18}O$ and $\delta^{13}C$ values in tree-ring subdivisions of *Pinus ponderosa* are best explained if

lags between the initial formation of tracheids and the production of cellulosic secondary cell walls are taken into account (Belmecheri et al., 2018). Hence, integrating the rate of xylogenesis with the understanding of isotope-climate relations is a novel avenue that can help improving the interpretation of stable isotopes in tree-ring records.

For tree-ring based climate reconstructions in general, and for identifying or eliminating divergences in particular, multi-proxy

and multi-site investigations represent good alternatives. The multi-proxy approach combines two or more records of different tree-ring variables (ring width, wood density, cellulose $\delta^{13}C$, $\delta^{18}O$ (or $\delta^2H$) values, cell wall thickness, tracheid-lumen diameter, or other wood anatomical traits) measured at the same site. Indeed, combining proxies with the same dominant control, but different secondary controls, tends to accentuate the common climate signal (McCarroll et al., 2011; Gennaretti et al., 2017; Wang et al., 2019). If a process generating divergence does not affect the tree-ring proxies in the same way, combining several

different proxy records can help reduce the amplitude of divergence between the climate signal and its reconstruction. However, applying the multi-proxy approach may not be straightforward as a specific climate parameter may generate responses of different frequencies for different proxies. Extracting and combining responses of common frequencies is required to produce a robust climate signal (see McCarroll et al., 2011 for a discussion). The multi-site approach consists in combining records from different sites (for instance a mesic and a xeric one: Lavergne et al., 2016). In addition, one can combine the

multi-site and multi-species approaches. For instance, such assemblages of isotopic records from the cellulose of *Fagus sylvatica*, *Quercus petraea*, *Abies alba*, *Picea abies*, *Pinus sylvestris* from several Swiss sites was shown to improve the reliability of temperature and precipitation reconstructions by cancelling out some biological noise (Saurer et al., 2008).

After going through the above options for limiting the climate-change induced isotopic divergences, the confrontation of

reconstructions from tree-ring isotopic series with independent archival systems, is a relevant approach for validating the produced reconstructions. The other archives potentially useable for such purpose include geopotential height, ice cores,



speleothems, lake sediments and historical data (Miller et al., 2006; Etien et al., 2008; Xu et al., 2016; Andreu-Hayles et al., 2017; Dinis et al., 2019; Muangsong et al., 2019; Yang et al., 2019).

### 4.3 Pollution effects

Trees are sensitive to changes in air quality and abundant literature depicts tree-ring isotopic series recording anthropogenic pollution stress or improvement of air quality (e.g., Savard, 2010; Thomas et al., 2013; Mathias and Thomas, 2018). In such cases, the isotopic series may display long-term departures from modelled unperturbed climatic trends, and an overall reduction of the tree-ring isotopic sensitivity to climatic conditions (Rinne et al., 2010; Doucet et al., 2012; Boettger et al., 2014; Savard et al., 2014). Cases of pollution effects on tree-ring $\delta^{13}C$ series abound in the literature (Table 1), but only rare studies on this 425 topic report significant effects on $\delta^{18}O$ (or $\delta^{2}H$) chronologies (Savard et al., 2005; Rinne et al., 2010; Boettger et al., 2014).

Pollution effects can occur in regions exposed to anthropogenic acidifying emissions, for which the effects possibly started with the earliest phase of industrialization, 170 years ago, but globally, the foremost burst of emissions occurred after the Second World War. The overall spatial and temporal extents of the pollution stress on plants closely relate to regional economic 430 developments in industrialised countries (e.g., de Vries et al., 2014), including mining, transformation industries, transportation using hydrocarbon combustion engines and power generation based on hydrocarbon burning. The type of emitters influences the spatial extent of the pollution footprint on tree-ring isotopic series. Large smelters and coal fired-power plants with high chimneys can affect downwind trees at more than 110 km (Savard et al., 2004) and even remote trees (150 km of more; Boettger et al., 2014), whereas emissions from highways show limited spatial reach due to the near ground level of the car 435 exhausts (Leonelli et al., 2012). Pollutants potentially detrimental to trees are sulphur dioxide ($SO_2$), nitrogen oxides (NOx), ozone ($O_3$), particulate matter (PM) and volatile organic C (VOC). The degree of effects on biological functions will differ with the type and intensity of exposures to air pollutants (chemistry, chronic or short exposures, and acute or mild levels).

The effects of ozone on trees' respiration, C assimilation, and stomatal conductance are complex (Matyssek et al., 2008; 440 Matyssek et al., 2010; Grulke and Heath, 2019), and dose-response models can help predict the extent of the reactions (Agathokleous et al., 2019). Beech and spruce trees exposed to elevated $O_3$ (and $pCO_2$) in greenhouse chambers show different isotopic sensitivity to pollutants with age, amongst the tested trees, juvenile beech trees being the most sensitive of these experimental trees (Grams et al., 2007). The impacts of $O_3$ mixed with other pollutants are difficult to predict in field conditions, and even more if hydric conditions and relative humidity are changing as well (Grulke and Heath, 2019). 445 Documented response mechanisms of trees exposed to chronic $SO_2$ emissions include changes in stomatal conductance, photosynthesis, dark respiration, starch production and priority of C allocation (Darrall, 1989; Meng et al., 1995; Kolb and Matyssek, 2001; Wagner and Wagner, 2006; Grams et al., 2007). With $SO_2$, exposition at 25 mg/m$^3$ induces photoinhibition and decreases A, g and Ci of plants (Duan et al., 2019). The exact mechanisms responsible for closing stomata differ between $SO_2$, $O_3$ and non-harmful $CO_2$ in terms of molecular biology. In fact, $O_3$- and $CO_2$ induce closure through similar mediating



genes as a protection mechanism but for stress avoidance and as a stimulating agent, respectively. $SO_2$ induces closure through guard cell mortality (Ooi et al., 2019). As seen in controlled experiments, NOx can have species-specific positive or negative effects on trees, opening of stomata, stimulating $CO_2$ assimilation and increasing biomass when beneficial (Siegwolf et al., 2001), and reducing g, A, or the root to shoot ratios when detrimental (Siegwolf et al., 2001; Hu et al., 2015; and references therein).


The response mechanisms behind the C and O isotopic fractionations in trees exposed to pollutants in controlled and field conditions are complex. When tested alone, deposition of NOx can either increase or decrease tree-ring $\delta^{13}C$ values (Siegwolf et al., 2001). $SO_2$ and $O_3$ may exert direct influences on $\delta^{13}C$ values in leaves, and sometimes indirectly on $\delta^{18}O$ values (e.g., Matyssek et al., 2010, Savard, 2010). $SO_2$, $O_3$ and NOx can also change extrinsic factors such as lowering rain pH and

increasing soil acidity, which can in turn modify the isotopic assimilation by trees (de Vries et al., 2014; Sensuła, 2015; Yang et al., 2018). For trees that grew under the influence of $SO_2$-dominated emissions from brick factories and coal mines, decreased g and increased dark respiration and production of starch generated long-term $\delta^{13}C$ increases but no significant $\delta^{18}O$ changes in pine and oak trees; starch having higher $\delta^{13}C$ values than cellulose (Wagner and Wagner, 2006; Rinne et al., 2010). In these cases, increased respiration rates expelled higher proportions of light C and generated tissues with higher $\delta^{13}C$ signals without

changing the $\delta^{18}O$ values (Kolb and Matyssek, 2001; Wagner and Wagner, 2006). In other contexts, lower g can also explain long-term $\delta^{13}C$ increases in coniferous trees growing near $SO_2$ sources (e.g., Martin and Sutherland, 1990; Savard et al., 2004; Rinne et al., 2010). Concomitant lower $\delta^{2}H$ (or $\delta^{18}O$) trends in the early phases of exposure to pollutants seem coherent with extrinsic factors regulating these $\delta^{2}H$ ($\delta^{18}O$) relationships inverse to the $\delta^{13}C$ trends, coeval with drastic decreases of $c_i$ (Savard et al., 2020). For instance, in a case of severe exposure of spruce trees to metal smelter emissions, acidification of upper soil

layers possibly induced water uptake by remaining efficient roots at depth into soils, where source water $\delta^{2}H$ values are low (Savard et al., 2005). In a case of urban diffuse emissions, similar mechanisms may explain the low-frequency inverse $\delta^{13}C$ and $\delta^{18}O$ patterns because they are synchronous with significantly low Ca/Mn ratios. Such conditions are not favouring an increase in photosynthetic rates (Doucet et al., 2012), the alternative to lowering g for explaining a $c_i$ decrease in the intrinsic dual-isotopic foliar responses (Scheidegger et al., 2000).


All the described response mechanisms to airborne emissions are long-term and independent of climatic effects, and they can diminish significantly the tree-ring isotopes-climate correlations (Rinne et al., 2010; Boettger et al., 2014; Savard et al., 2020). As a result, pollution-influenced isotopic series enclose divergences with climatic records. Hence, using such series to develop response functions will predictably generate reconstructed climatic series departing from true past climate.



### 4.4 Ways to avoid isotopic divergences due to pollution


Only few studies have reported changes in the relationships between climatic conditions and tree-ring isotopes due to air pollutants (Table 1), and incited to careful consideration of this type of divergences prior to attempting reconstruction of climate in a given region. The industrial time during which pollution effects could disturb the climate-isotope relationships overlaps with the periods of instrumented meteorological measurements. Interestingly, quantification of $SO_2$ effects on ring

$\delta^{13}C$ series from trees growing in field conditions reveals the complex and vain task of attempting to correct for such divergences (Rinne et al., 2010). Therefore, to avoid erroneous reconstruction using isotopic series biased due to pollution requires removal of the part of the series corresponding to the divergence period prior to carrying out the climate-isotope calibration. This remedy may apply for regions where instrumented meteorological series are much longer than the period of disturbance, restricting the length of the tree-ring suites for isotope-climate calibration. In regions where the divergence period

is too long to allow for proper statistical calibration, the only remedy is to explore for remote stands non-exposed to long-distance airborne pollutants. Hence, tree-ring isotopic series used for climatic reconstruction need asserting that they are devoid of pollution effects, or truncating so to keep only the unaffected tree-ring segments.

Currently, mechanistic tree-growth or proxy system models integrating isotopic results can account for several non-climatic

factors (Guiot et al., 2014), but none takes into account the physiological reactions to degradation of air quality, which would open possibilities to dodge this type of isotopic-climatic departures from stationarity. Attempting to eliminate divergences due to pollution may perhaps proceed with the multi-proxy and multi-site approaches in the future (see Section 4.2). Clearly, at this stage of development, the most efficient way to circumvent the potential isotopic divergences between climatic parameters and isotopic series and the inconvenience of short calibration segments due to airborne pollutants is to select trees outside

pollutant deposition zones.

In summary, caution should prevail when investigating trees from stands in peripheral areas of large cities, heavy industrial centres or major point sources. Emissions from such zones may have altered the tree-ring isotopic responses and the sensitivity of these proxies to climate, with potential strong effects after 1945, contemporaneous with latent $pCO_2$ and climate-change

divergences, during the calibration period for climatic reconstruction. Screening isotopic series for pollution effects and circumventing the related isotope-climate divergences are required steps to produce valid climatic reconstruction.

### 5 The isotopic divergence problem – Perspective

The sensitivity of trees to changes in their environment imparts at the same time the strengths and weakness of the tree-ring isotopic proxies for climatic reconstruction, spurring specific proxies to record climate variations and respond to multiple

triggers with varying dominance through time. Anthropogenic climate change seem to cause the observed recent disconnections between climatic parameters and isotopic variables. At longer time scale, switching of climatic regimes and





climate control on trees may be at the origin of observed isotopic divergences. The reported cases of recent isotopic divergences show that foliar physiological responses to rising $pCO_2$ and acidifying pollutants generate lower effects on $\delta^{18}O$ values than on $\delta^{13}C$ values, and therefore suggest that $\delta^{18}O$ data constitute a more appropriate proxy for reconstructing climate using

statistical approaches. However, recent studies report also climate-$\delta^{18}O$ divergences due to climate change, although of lower amplitudes than those documented for $\delta^{13}C$ series (Table 1). In this sense, dendroisotopists should acknowledge and fully seize the importance of isotopes-climate divergences.

Following the same evolution path of recognizing tree-ring statistical $\delta^{13}C$ divergence caused by the rising-$pCO_2$ physiological

effects and correcting $\delta^{13}C$ series before climatic reconstruction, isotopic divergences caused by climate change and pollution need routine testing, and handling when identified (Table 2). In studies conducted recently, scientists are addressing the issue prior to digging into a specific climatic question. For instance, early wood and late wood cross-correlations of $\delta^{13}C$ and $\delta^{18}O$ records in southwestern USA pine trees revealed low frequency modes (divergence) due to climate change (Table 1). The statistical removal of these multi-year cross correlations helped improve the relationships between isotopes and seasonal

climatic data explored afterwards using a mechanistic isotope-climate forward model. The model successfully predicted different rain patterns from unimodal to bimodal precipitation from North to South (Szejner et al., 2018; Szejner et al., 2019).

Indeed, process-based isotopic models can serve pinpointing causes of divergence over the last century, when measured meteorological data are available. Depending on the sophistication of the mechanistic or proxy-system approaches, modeling

may even compensate for divergences due to climate change. The tree-ring isotopic outputs from process-based models are sensitive to changes in key input parameters, such as $\delta^{18}O$ values in rain, vapour and soil water (Lavergne et al., 2017). Therefore, for this exact reason, inversing mechanistic models may not completely prevent producing flawed reconstruction of key parameters if changes in climatic regimes occurred during the reconstructed period. In this sense, practitioners should keep in mind that interplays of key parameters might change through time due to modulations by distant forcing. An example

of such cases, as mentioned in Section 4.1.1, is that the assumed joined changes of tree-ring $\delta^{13}C$ values and temperature was altered when the relationship between solar radiation (cloudiness) and temperature changed with cloud circulation triggered by Arctic Oscillations during the last 500 years in Scandinavia (Young et al., 2010).

For dealing with isotopic divergences, again, scientists will need to adopt approaches with the level of complexity suiting their

main goals. Empirical statistical modeling is practical and readily applicable, but limited in its skill to reproduce the multiple physiological constraints of natural conditions. For that reason, statistical modelling may produce climatic reconstructions of variable reliability. Ecophysiological mechanistic modeling is labour intensive and theoretically powerful at the same time. In a reported comparison of the two techniques, statistical reconstructions provided precipitation and temperature series similar to the process-based inversed reconstructions (Boucher et al., 2014), owing to the strength of the statistical correlations between




precipitation, temperature and the isotopic series. However, the undeniable finesse of the ecophysiological inversion comes from its incorporation of multiple proxies (ring width, $\delta^{13}C$ and $\delta^{18}O$) reflecting simultaneously several processes controlling them (Guiot et al., 2014). Another gain when working with mechanistic modeling is in getting deeper knowledge on processes controlling tree-ring properties. Nevertheless, not all model inputs (daily climatic parameters and rain $\delta^{18}O$) are readily available or derivable, and if available, daily records may not cover the entire calibration period (Section 2.2). Importantly, not all mechanistic models have the same readiness for climatic inversion. MAIDENiso is the only model that was successfully inversed for reconstructing climate using several proxies including tree-ring isotopes, in three regions of the world (France, Québec and southern Argentina; Danis et al., 2012; Boucher et al., 2014; Lavergne et al., 2017). Moreover, ecophysiological models and experts with the required coding knowledge for improving the existing executable formats are rare. In addition, the complexity with process-based models resides in parametrisation prior to climatic reconstruction. Therefore, depending on the availability of information, many practitioners will decide upon using statistical transfer functions and still produce very informative climatic series. In the long term, increasing the number of reconstructions worldwide, whatever the employed approach, will tend to reduce gaps and errors in reconstructed series.

The way forward in solving the isotopic divergence issue may partly come from conducting research programs using inverse process-based models simulating simultaneously meteorology, plant-life mechanisms, and other non-climatic factors. In fact, process-based models continuously evolve with the addition of modules to expand their abilities. One can imagine developing and implementing modules for pollution and solar radiation that would help address some of the important issues underlined here. In closing, our main message is that reconstruction of global climatic patterns requires addressing the drawback of isotopic divergences, and the means to leap beyond this challenge reside in joining efforts from vegetation biologists, ecophysiologists, modellers, dendrochronologists, isotopists, statisticians and paleoclimatologists.

**Acknowledgements**

NSERC (Persistence project), and Environmental Geoscience and Climate Change programs for financial support. Dr Étienne Boucher for stimulating discussions. Dr. Lauriane Dinis for helpful comments on an early version of the manuscript, and Dr. Soumaya Belmecheri for a thorough and constructive review of the article. NRCan contribution number: 20190529.

**Author contribution**

MMS and VD: joint conception of outline and literature review. MMS: original draft and visualization; VD: several sections and review.





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





**Tables and Figures**

**Table 1. Reported critical divergences of correlations between isotopic results and instrumental climatic series (other than sampling, stand dynamics and juvenile effects).**

| Isotopes | Climate Parameters | Tree species | Causes | Region | Author(s) |
|---|---|---|---|---|---|
| $\delta^{13}C$ | Summer T | *Quercus robur* | CC: longer growth season | Eastern England | Aykroyd et al., 2001 |
| $\delta^{13}C$, $\delta^{18}O$ | Summer T, Pc | *Quercus petraea; Pinus sylvestris* | CC: physiological adaptation to higher T, change in moisture origin | Switzerland | Reynolds et al., 2007 |
| $\delta^{13}C$ | Summer T | *Pinus sylvestris* | CC: earlier Summer | Eastern Finland | Hilasvuori et al., 2009 |
| $\delta^{13}C$ | Summer T& Pc | *Quercus robur, Pinus sylvestris* | Poll: $SO_2$ from close emitter | Southeastern England | Rinne et al., 2010 |
| $\delta^{13}C$ | Summer cloud cover, T | *Pinus sylvestris* | CC: AO, decoupling of T and radiations | Northwestern Norway | Young et al., 2010 |
| $\delta^{13}C$, $\delta^{18}O$ | Tmax, RH | *Larix decidua* | CC: drier climate; deeper soil water | French Alps | Daux et al., 2011 |
| $\delta^{13}C$, $\delta^{18}O$ | Summer T & Pc | *Pinus sylvestris* | CC: change in T, irradiance & cloud circul. | N. boreal zone | Seftigen et al., 2011 |
| $\delta^{13}C$ | Summer T& Pc | *Larix decidua* | Poll: traffic/vehicles | Italian Alps | Leonelli et al., 2012 |
| $\delta^{18}O$ | Summer Pc | Pinus halepensis | CC: increase of drought; deeper soil water | Greece | Sarris et al., 2013 |
| $\delta^{13}C$ | *No link* | *Juniperus virgiana* | Poll: distant $SO_2$ emitters | Appalachians, USA | Thomas et al., 2013 |
| $\delta^{2}H$, $\delta^{13}C$, $\delta^{18}O$ | RH | *Abies alba* | Poll: distant $SO_2$ emitters | Southwestern Germany | Boettger et al., 2014 |
| $\delta^{13}C$ | RH, T | *Abies georgei* | CC: water stress | Western China | Liu et al., 2014 |
| $\delta^{13}C$ | Tmax | *Picea mariana & glauca* | Poll: oil sands mining operations | Alberta, Canada | Savard et al., 2014 |
| $\delta^{13}C$, $\delta^{18}O$ | Summer T Spring Pc | *Picea mariana* | CC; NAO longer growth season | Northeastern Canada | Naulier et al., 2015b |
| $\delta^{13}C$ | Spring-Sum. T | *Sabina przewalskii* | CC: change in cloud circulation | Tibet | Wang et al., 2016; 2019 |
| $\delta^{13}C$, $\delta^{18}O$ | VPD | *Pinus ponderosa* | CC: increase of drought | Southw. USA | Szejner et al., 2018 |
| $\delta^{18}O$ | Spring AO, spring NAO | *Cryptomeria japonica* | CC: spring AO-EASM changes | Northeastern Japan | Sakashita et al., 2018 |
| $\delta^{13}C$ | *No link* | *Picea rubens* | Poll: distant $SO_2$ emitters | Appalachians, USA | Mathias & Thomas, 2018 |
| $\delta^{18}O$ | May-July T, RH, PDSI | *Abies forrestii* | CC: change in moisture origin | Southwestern China | An et al., 2019 |
| $\delta^{13}C$ (WUE) | Summer Tmax | *Picea mariana & glauca* | Poll: bitumen mining, metal smelter emissions, global $CO_2$ rise | Alberta & Québec, Canada | Savard et al., 2020 |

T : temperature. Tmax : maximum temperature. RH : relative humidity. Pc: precipitation. VPD: vapour pressure deficit. AO : Arctic oscillations. NAO : North Atlantic Oscillations. PDSI: Palmer drought severity index. CC : climate change. Poll : pollution stress. EASM: East Asian summer monsoon.



**Table 2. Summary of isotopic divergences and suggested corrective measures to use prior to climatic reconstruction.**

| Divergence type | Corrective/preventive measures |
|---|---|
| *Sampling and data-treatment artefacts* | Mathematical modifications (correction factor, average adjustment) |
| | Analysis of several stem cores from a large number of trees |
| *Stand dynamics* | Truncation of affected early part of the isotopic series |
| | Mathematical removal of long trends (ex. regional curve standardization) |
| *Effects of rising $CO_2$* | Ecophysiological modeling |
| | Pre-industrial correction |
| *Climate change* | Truncation of divergent part of the isotopic series for calibration |
| | Usage of late wood isotopic signals (mostly for deciduous trees) |
| | Combination of several tree-ring proxies, species or sites |
| | Validation using independent archival systems |
| *Pollution* | Truncation of tree rings corresponding to period of effects |
| | Selection of stands outside the influence area of anthropogenic emissions |



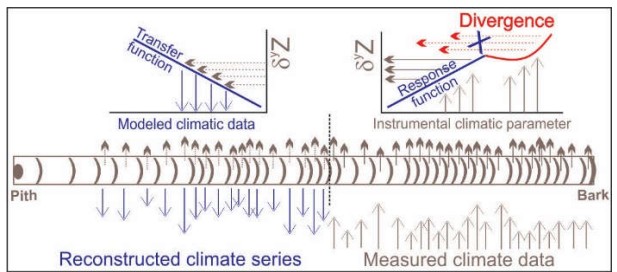

**Figure 1: Schematic representation of a tree-ring isotope series used for climatic reconstruction. The statistical correlations between a climate parameter and a tree-ring isotopic proxy $\delta^y Z$ leads to the development of the response and transfer functions used to model climate retrospectively. Prior to reconstruction, corrective measures addressing the isotopic divergence are required.**