# Peer review of "An Overview on Isotopic Divergences – Causes for instability of Tree-Ring Isotopes and Climate Correlations"

_Climate of the Past, 2020_

## Referee Comment (RC1) · Anonymous Referee #1 · 12 Mar 2020

The manuscript reviews recent development of isotopic dendroclimatology, addressing possible divergence problem in tree-ring d13C and d18O. In my opinion, this kind work is very important when isotopic dendroclimatology has been paid more attention and plays more important role in high-resolution paleoclimate reconstruction. However, the current manuscript should be reorganized and concentrated in d13C. Physiological mechanisms between tree-ring d13C and d18O are quite different, therefore, comparing with tree-ring d13C, tree-ring d18O did not show recognizable effects from rising pCO2 (Lines 240-241) and pollution (Lines 424-425). Just as the authors said in Lines 514-515, tree-ring d18O is a more appropriate proxy for climate reconstruction. And, the phenomenon of divergence between the d18O and climate is really few worldwide.

I also suggest the authors adding one section discussing uncertainty. Isotopic dendroclimatology is a subject based on chemical experiment. Unlike tree-ring width or density, the result of tree-ring d13C or d18O measurements are different to verified again from their core or disk samples, due to time consuming and great expense. It is possible to introduce mistakes during many steps of experiments, for example, impure cellulose and unreliable measurements caused by bad condition of the isotope ratio mass spectrometer. Some other uncertainties also exist. First is sampling strategy. We should understand what kind of tree could be used for climate reconstruction. As recommended by classical "The principle of limiting factors" (Fritts 1976), site selection is very important when one would employ trees to infer climate change. It is also important to isotopic dendroclimatology. Because mixed (deep phreatic water, shallow ground water, precipitation...) ground water may disturb tree-ring d18O (A tree in flowing figure), tree-ring d18O of B tree only absorb precipitation. Although cellulose d13C and d18O could be measured for any tree from any site, but for purpose of climate correlation, it should be carefully selected.

FIG. 1.5. Trees growing on sites where climate seldom limits growth processes produce rings that are uniformly wide (A). The ring widths provide little or no record of variations in climate and are termed *complacent*. Trees growing on sites where climatic factors are frequently limiting produce rings that vary in width from year to year depending upon how severely limiting climate has been to growth (B). These are termed *sensitive*.

Second uncertainty may be introduced by different samples for measurement (extractive-free samples,  $\alpha$ -cellulose, whole wood or holocellulose). And, for different chemical extraction methods (Green's method, Brendel's method...). Third uncertainty may be introduced by "pooling" or "not pooling".

Special comments are as follow.

Lines 13-19, changes on physiology ( $f_0$  and peclet effect...) should be mentioned here.

Line 32, need a reference

Lines 38-32, it is no need to descript growth divergence

Line 72, "concentration" is better than "pressure", also in Line 236

Lines 94-95, need references

Line 157, 5 year is not enough

Lines 175-177, one advantage for tree-ring isotope chronology is no need to detrending. If detrending for the isotope chronology, some climate signals may be lost.

Line 263, "1850s" is easy to understanding than "last 170 years"

Line 302, relative humidity and RH, repeat

Line 376, tropic/

Lines 399-412, does multi-proxy approach introduces more climate noise?

Section 3.3. there is only one sentence to state the situation of cellulose

d180 (Line 240). I recommend to delete d180 discussion in this manuscript. In addition, removing effect of increasing CO2 from the d13C series has been discussed in many literatures. Please shorten this section. Table 1, check which one use pooling method.

---

## Referee Comment (RC2) · Anonymous Referee #2 · 1 May 2020

Martine M. Savard and Valérie Daux

**Anonymous Referee #2**

First, I would like to thank the authors for conducting this synthesis. The authors did a great job on synthesizing and explaining all the different sources of divergence caused by multiple factors recorded in Carbon and Oxygen isotopes in the wood.

At the beginning of this review, the authors explain the differences between the "divergence" topic in the tree ring community versus the divergence that can be found in isotopic measurements in Tree rings. I see why the Authors are attributing the term divergence to the examples they show. However, it is not clear if the term divergence is the correct term. It is fine to use this term as long the authors make sure that they are referring to the divergence to the climate signal and eventually highlight that this "issue" falls into the problems we as scientists have when we want to interpret the isotopic records in Tree rings. I do appreciate the sections where they make recommendations and a strong call to the good practices so future researchers can take this advice to minimize the chances of losing the climatic signal.

---

## Referee Comment (RC3) · Anonymous Referee #2 · 1 May 2020

One comment I should mention is that the review is highly focused on climate reconstructions, while the ecophysiological responses to environmental cues are somewhat left a little bit on the side, as something that is dampening or disrupting the climate signal.

So, the more specific comments are more targeted to references in the literature (given this manuscript is a review) plus some other clarifications if the author agrees.

Line 125. The Model "MAIDEN" is not well explained, so I recommend explaining it a little bit, so the reader can understand what the model it's all about.

Line 131 The citation for the Vaganov model it should be correctly cited, or add the

papers where Vaganov published originally, then, of course, you can use other citations as usage examples.

Line 89 and Line 359 The percentage of oxygen isotope exchange during cellulose synthesis, as you mention, can indeed be variable. Recently there is a published paper addressing this same possibility and highlights some of the possible hypotheses that can be involved in such phenomena. Probably this is a reference you might be interested in exploring. New Phytologist (2020) doi: 10.1111/nph.16484

Line 253 The PIN correction of the pCO2 influence on the D13C discrimination should be double-checked. I think Gagen et al. 2007 made the first mention of the Pin correction that I know of. The Holocene, 17(4), 435–446. https://doi.org/10.1177/0959683607077012

Line 206 Another recent publication Citation that you might be interested in exploring about age effects in Tree ring isotopes is form Xu et al. 2020. I think this is relevant to your review as it addresses the age-related effect concerning Climate reconstructions. 2020 Journal of Geophysical Research: Biogeosciences, 0–2. https://doi.org/10.1029/2019JG005513

Lien 262. I agree that there is no overarching consensus over how to correct the pCO2 effects on the discrimination of 13C. But I find a bit troubling this sentence "A wise approach is to test the various corrective methods and assess the performance of the resulting series with climatic reconstruction model." This statement is for me, suggesting that we should select the best fit to climate. I think this is a bit biased and undermined the fact that we still do not fully understand how the pCO2 is affecting gs and A. so I think this part needs to be careful on not incentivize researchers to select the best fit, but instead, incentive to investigate what is the mechanisms and how the pCO2 is or not affecting the Carbon chronologies. Then I suggest reviewing Global Change Biology, 22(2), 889–902. https://doi.org/10.1111/gcb.13102

Line 287 I think this part needs this reference. Dorado-Liñán, I et al. 2016. Climate

Dynamics, 47(3–4), 937–950. https://doi.org/10.1007/s00382-015-2881-x

Line 345 This reference also can be useful here Carbone, M. S. et al. 2013, The New Phytologist, 200(4), 1145–55. https://doi.org/10.1111/nph.12448

---

## Author Comment (AC1) · 6 May 2020

General comments Reply – We would like to thank referee 1 for the suggestions made during this review.

The manuscript reviews recent development of isotopic dendroclimatology, addressing possible divergence problem in tree-ring d13C and d18O. In my opinion, this kind work is very important when isotopic dendroclimatology has been paid more attention and plays more important role in high-resolution paleoclimate reconstruction. However, the current manuscript should be reorganized and concentrated in d13C. Physiological mechanisms between tree-ring d13C and d18O are quite different, therefore, comparing with tree-ring d13C, tree-ring d18O did not show recognizable effects from rising pCO2 (Lines 240-241) and pollution (Lines 424-425). Just as the authors said in Lines 514-515, tree-ring d18O is a more appropriate proxy for climate reconstruction. And, the phenomenon of divergence between the d18O and climate is really few worldwide.

REPLY - Regarding presenting the review on '18O series, indeed, rising pCO2 does not create divergence with climate for the '18O series, as we pertinently explain, but other causes do: change in climatic regimes and pollution (sections 4.1 and 4.3). That is why we judge pertinent keeping the d18O series in this review (see Table 1).

I also suggest the authors adding one section discussing uncertainty. Isotopic dendroclimatology is a subject based on chemical experiment. Unlike tree-ring width or density, the result of tree-ring d13C or d18O measurements are different to verified again from their core or disk samples, due to time consuming and great expense. It is possible to introduce mistakes during many steps of experiments, for example, impure cellulose and unreliable measurements caused by bad condition of the isotope ratio mass spectrometer.

REPLY - Uncertainties do exist for any kind of physical measurement, including tree ring width or density determination. We agree that in isotopic dendroclimatology, the chemical extraction of cellulose and the spectrometric measurements are critical steps. Impure cellulose and unreliable measurements yield bad data, which indeed more than likely diverge from climate. We can introduce a sentence of caution and refer to several papers devoted to good analytical practices (for instance: Loader et al., 1997; Boettger et al., 2007; Wieloch et al., 2011; Kagawa et al., 2015; Andre-Hayles et al., 2019) but we do not think this subject has to be extensively discussed in this paper. Andreu-Hayles, L., Levesque, M., Martin-Benito, D., Huang, W., Harris, R., Oelkers, R., Leland, C., Martin-Fernández, J., Anchukaitis, K.J., Helle, G., 2019. A high yield cellulose extraction system for small whole wood samples and dual measurement of carbon and oxygen stable isotopes. Chem. Geol. 504, 53–65. https://doi.org/10.1016/j.chemgeo.2018.09.007 Boettger, T., Haupt, M., Knöller, K.,

Weise, S.M., Waterhouse, J.S., Rinne, K.T., Loader, N.J., Sonninen, E., Jungner, H., Masson-Delmotte, V., Stievenard, M., Guillemin, M.T., Pierre, M., Pazdur, A., Leuenberger, M., Filot, M., Saurer, M., Reynolds, C.E., Helle, G., Schleser, G.H., 2007. Wood cellulose preparation methods and mass spectrometric analyses of $\delta$13C, $\delta$18O, and nonexchangeable $\delta$2H values in cellulose, sugar, and starch: An interlaboratory comparison. Anal. Chem. 79, 4603–4612. https://doi.org/10.1021/ac0700023 Loader, N.J., Robertson, I., Barker, a. C., Switsur, V.R., Waterhouse, J.S., 1997. An improved technique for the batch processing of small wholewood samples to $\alpha$-cellulose. Chem. Geol. 136, 313–317. https://doi.org/10.1016/S0009-2541(96)00133-7 Kagawa, A., Sano, M., Nakatsuka, T., Ikeda, T., Kubo, S., 2015. An optimized method for stable isotope analysis of tree rings by extracting cellulose directly from cross-sectional laths. Chem. Geol. 393–394, 16–25. https://doi.org/10.1016/j.chemgeo.2014.11.019 Wieloch, T., Helle, G., Heinrich, I., Voigt, M., Schyma, P., 2011. A novel device for batchwise isolation of $\alpha$-cellulose from small-amount wholewood samples. Dendrochronologia 29, 115–117. https://doi.org/10.1016/j.dendro.2010.08.008

Some other uncertainties also exist. First is sampling strategy. We should understand what kind of tree could be used for climate reconstruction. As recommended by classical "The principle of limiting factors" (Fritts 1976), site selection is very important when one would employ trees to infer climate change. It is also important to isotopic dendroclimatology. Because mixed (deep phreatic water, shallow ground water, precipitation...) ground water may disturb tree-ring d18O (A tree in flowing figure), tree-ring d18O of B tree only absorb precipitation. Although cellulose d13C and d18O could be measured for any tree from any site, but for purpose of climate correlation, it should be carefully selected.

REPLY- Thank you for raising that up. Indeed site selection is a crucial step in paleoclimate research. If trees and/or sites are not well selected, one of the main risks is that their '18O and/or '13C isotopic series show no significant relation with climate. In that case, reconstruction is not possible. Therefore, we judge that using poor criteria

for site selection does not have to be dealt with in this paper as we assume that the readership is well aware of the initial step in adequate site selection.

Second uncertainty may be introduced by different samples for measurement (extractive-free samples, $\alpha$-cellulose, whole wood or holocellulose). And, for different chemical extraction methods (Green's method, Brendel's method. . .). Third uncertainty may be introduced by "pooling" or "not pooling".

REPLY- Different extraction methods, as well as measurements produced in different laboratories with different spectrometers and procedures, should produce comparable results. There are very few inter-lab calibration experiments. To our knowledge the only study dealing with such a comparison was produced by the ISONET group (Boettger et al., 2007). Another one is in progress in the frame of the THEMES project conducted by one of the two authors (Daux, Andreu-Hayles et al., in progress). These inter-laboratories comparisons show that shifts between isotopic levels exist between laboratories (high correlations but different absolute value) due to different extraction methods, use of different reference materials, different apparatus, etc. However, as long as the data included in an isotopic chronology have all been produced following the same protocols, by the same experimentalists, the data are consistent with one another and if they diverge from climate variations, the cause should be sought elsewhere.

Special comments are as follow.

Lines 13-19, changes on physiology (fo and peclet effect. . .) should be mentioned here.

REPLY- These items are covered in section 2.1. Here we cite the main CAUSES for divergences, not the mechanisms through which they operate. No change to these lines.

Line 32, need a reference

REPLY- We will add D'Arrigo et al., 2008.

Lines 38-32, it is no need to descript growth divergence REPLY- The referee probably means lines 38-42. Here we just present some background using growth divergence. No change to these lines.

Line 72, "concentration" is better than "pressure" , also in Line 236

REPLY- Atmospheric CO2 Pressure or pCO2 are well accepted and widely used. No change to these lines.

Lines 94-95, need references

REPLY- The references covering this topic are through equations 1-3, for which the citations are in the previous text. No change to these lines.

Line 157, 5 year is not enough

REPLY- Understood. The parenthesis underlines the fact that some researcher may opt for longer overlaps. No change to this line.

Lines 175-177, one advantage for tree-ring isotope chronology is no need to detrending. If detrending for the isotope chronology, some climate signals may be lost.

REPLY- We agree. That is exactly what the text explains. No change to these lines.

Line 263, "1850s" is easy to understanding than "last 170 years"

REPLY- We will modify the sentence from Âń. . . of rising pCO2 of the last 170 yearsÂż to: . . . of rising pCO2 since 1850.

Line 302, relative humidity and RH, repeat

REPLY- we just present the abbreviation (RH) for relative humidity here. No change to this line.

Line 376, tropic/

REPLY- Tropical is a well accepted English adjective. No change to this line.

**CPD**

Lines 399-412 – multi-proxy approach and more climate noise?

REPLY- We do not understand what the reviewer means. We say the contrary un the text: 'Indeed, combining proxies with the same dominant control, but different secondary controls, tends to accentuate the common climate signal'.

Section 3.3. there is only one sentence to state the situation of cellulose d18O (Line 240). I recommend to delete d18O discussion in this manuscript. In addition, removing effect of increasing CO2 from the d13C series has been discussed in many literatures. Please shorten this section.

REPLY- It is true that the literature raised the issue abundantly, but there is still no consensus on how to approach and correct the problem. This manuscript designed to be a review article should cover the matter and section 3.3 intend to do just that. Concerning the effects on ïĄd'18O values, it is worth explaining which articles address the potential pCO2 effects, even if nil or minimal. In addition, referee 2 pertinently suggests to integrate new references to this section. So we decide not to shorten the section.

Table 1, check which one use pooling method.

REPLY- good point. We indicated the studies using pooled series (asterisks in Table 1 that we can provide by means to be indicated by the editorial team), without identifying any specific common factors. Note that many of these studies validated that the use of pooled trees gave similar results to treatments of individual trees later averaged mathematically.

---

## Author Comment (AC2) · 6 May 2020

Please see main reply to reviewer #2

———————————————————

---

## Author Comment (AC3) · 6 May 2020

First, I would like to thank the authors for conducting this synthesis. The authors did a great job on synthesizing and explaining all the different sources of divergence caused by multiple factors recorded in Carbon and Oxygen isotopes in the wood.

At the beginning of this review, the authors explain the differences between the "divergence" topic in the tree ring community versus the divergence that can be found in isotopic measurements in Tree rings. I see why the Authors are attributing the term divergence to the examples they show. However, it is not clear if the term divergence is the correct term. It is fine to use this term as long the authors make sure that they are

referring to the divergence to the climate signal and eventually highlight that this "issue" falls into the problems we as scientists have when we want to interpret the isotopic records in Tree rings. I do appreciate the sections where they make recommendations and a strong call to the good practices so future researchers can take this advice to minimize the chances of losing the climatic signal.

REPLY – We sincerely thank referee 1 for the constructive comments and suggestions compiled above and below. Regarding the usage of 'divergence', we agree with the referee that this term should be restricted to describing tree-ring isotopic departures from climatic parameters. That is what we rigorously do in the manuscript. The introduction explains lines 30-32: ÂńWhen correlations between climatic parameters and tree-ring proxies show periods of instability such that correlations weaken, become non-significant or change in signs, the relationship between proxies and climatic data shows a 'divergence'. Âż Further down (lines 44-45): ÂńThe present article deals with the 'isotopic divergence', which we define here as the middle- to long-term (>10 years) loss or change in signs of correlations between a climatic parameter and tree-ring isotopic ratios (d13C, d18O, or rarely d2H). Âż

Anonymous Referee #2 Some other suggestions

One comment I should mention is that the review is highly focused on climate reconstructions, while the ecophysiological responses to environmental cues are somewhat left a little bit on the side, as something that is dampening or disrupting the climate signal.

REPLY – It is right to reckon that the article focuses on the tree-ring isotopes-climate relationships with the main purpose of climatic reconstruction as explained in the introduction (lines 61-63): ÂńGiven the need for careful assessments of isotopes as climate proxies for various regional contexts and tree species, this synthesis of the up-to-date information on isotopic divergences aims at: (1) describing the main isotopic divergence types and discussing their potential causes, and (2) reviewing research avenues to identify them and account for them (Table 2). Âż On one hand, wide ecological changes are not included here on purpose as we wanted to restrict the covered topic to TR isotopic divergences due to direct tree responses. On the other hand, we see the ecophysiological approaches to assess tree responses to changes as part of eventual solutions for circumventing some isotopic divergence issues (see for instance Section 5; lines 543-557). Therefore, we do not take action in response to this comment.

So, the more specific comments are more targeted to references in the literature (given this manuscript is a review) plus some other clarifications if the author agrees.

Line 125. The Model "MAIDEN" is not well explained, so I recommend explaining it a little bit, so the reader can understand what the model it's all about.

REPLY – Lines 121-125 define the general approach to mechanistic modeling, which applies to MAIDEN as well as to the other models of the kind. We do not want to place too much emphasis on MAIDEN, but following the suggestion of referee 1, we explain briefly the main structure of MAIDENiso as follows (starting lines 123): Âń Most models make forward predictions and allow verifying that the measured tree-ring isotopic trends compare well with the isotopic outputs modelled with the meteorological and non-meteorological inputs, and identifying processes behind isotopic responses. For instance, MAIDENiso is an expanded growth model which includes C and O modules. The model allows reproducing fractionation of carbon isotopes due to atmospheric $CO_2$ diffusion to the site of carboxylation, enzymatic photosynthesis and respiration, and estimates oxygen isotopes in precipitation, soil water and xylem water, and the fractionation in leaves due to due evapotranspiration and to biochemical formation of cellulose (details in Danis et al., 2012; Boucher et al., 2014). Âż

Line 131 The citation for the Vaganov model it should be correctly cited, or add the papers where Vaganov published originally, then, of course, you can use other citations as usage examples.

REPLY – Good point. We will correct the name of the model and will refer to Vaganov

et al., 2011.

Line 89 and Line 359 The percentage of oxygen isotope exchange during cellulose synthesis, as you mention, can indeed be variable. Recently there is a published paper addressing this same possibility and highlights some of the possible hypotheses that can be involved in such phenomena. Probably this is a reference you might be interested in exploring. New Phytologist (2020) doi: 10.1111/nph.16484

REPLY – Good point; this newly published reference is pertinent. We will add a citation to this article at former line 359 and add the full reference to the final list. The text will read: Âń However, this proportion may vary over growing seasons and longer periods, and due to relative humidity conditions, (Gessler et al., 2009; Szejner et al., 2020). Âż

Line 253 The PIN correction of the pCO2 influence on the D13C discrimination should be double-checked. I think Gagen et al. 2007 made the first mention of the Pin correction that I know of. The Holocene, 17(4), 435–446. https://doi.org/10.1177/0959683607077012

REPLY – Correct. We will replace the citation to McCarroll et al., 2009 by a citation to Gagen et al., 2007.

Line 206 Another recent publication Citation that you might be interested in exploring about age effects in Tree ring isotopes is form Xu et al. 2020. I think this is relevant to your review as it addresses the age-related effect concerning Climate reconstructions. 2020 Journal of Geophysical Research: Biogeosciences, 0–2. https://doi.org/10.1029/2019JG005513

REPLY – Good point; this freshly appeared reference is pertinent. We will add the text below with the citation to this article at the end of the last paragraph in section 3.3, and add the full reference to the final list. Âń Finally, in some cases, though there is no trend in the tree-ring isotopic series, the response to climate in the isotopic chronologies may be age-dependent. For instance, in Picea Schrenkiana from northwestern

China, d18O and d13C values in trees under 125 years have stronger response to relative humidity than trees older than 270 years. The diminution of the strength of the correlations with tree age advocates for the incorporation of young trees only to develop a non-divergent composite chronology. Âż Xu, G., Wu, G., Liu, X., Chen, T., Wang, B., Hudson, A., 2020. Age-related climate response of tree-ring d13 C and d18 O from spruce in northwestern China, with implications for relative humidity reconstructions 0–2. https://doi.org/10.1029/2019JG005513 Lien 262. I agree that there is no overarching consensus over how to correct the pCO2 effects on the discrimination of 13C. But I find a bit troubling this sentence "A wise approach is to test the various corrective methods and assess the performance of the resulting series with climatic reconstruction model." This statement is for me, suggesting that we should select the best fit to climate. I think this is a bit biased and undermined the fact that we still do not fully understand how the pCO2 is affecting gs and A. so I think this part needs to be careful on not incentivize researchers to select the best fit, but instead, incentive to investigate what is the mechanisms and how the pCO2 is or not affecting the Carbon chronologies. Then I suggest reviewing Global Change Biology, 22(2), 889–902. https://doi.org/10.1111/gcb.13102

REPLY – We agree. The text will be modified as follows: Âń A wise approach is to investigate the potential influence of pCO2 on isotopic ring series and the gas-exchange response mechanisms in trees prior to selecting a corrective method (Voelker et al., 2016; Savard et al., 2020). Âż

Line 287 I think this part needs this reference. Dorado-Liñán, I et al. 2016. Climate Dynamics, 47(3–4), 937–950. https://doi.org/10.1007/s00382-015-2881-x

REPLY – We agree. The modified text will be: Âń Yet, studies in northwestern Norway (Young et al., 2010), the Northern boreal zone (Seftigen et al., 2011) and Northern Spain (Dorado-Liñan et al., 2016) depicted divergences between temperature records and 13C series of pines (Pinus sylvestris or Pinus uncinata) during episodes of decoupling between irradiance and temperature linked to either changes in large scale

atmospheric circulation (in the first two references), or large volcanic eruptions (in the third one). Âż Dorado-Liñán, I., Sanchez-Lorenzo, A., Gutierrez Merino, E., Planells, O., Heinrich, I., Helle, G., Zorita, E., 2016. Changes in surface solar radiation in Northeastern Spain over the past six centuries recorded by tree ring $\delta$13C. Clim. Dyn. 47, 937–950. https://doi.org/10.1007/s00382-015-2881-x

Line 345 This reference also can be useful here Carbone, M. S. et al. 2013, The New Phytologist, 200(4), 1145–55. https://doi.org/10.1111/nph.12448

REPLY – We agree. The modified text will be: ÂńThe proportion of direct assimilates increases progressively at the expense of reconverted stored material, until they are the only carbohydrate source for building new plant tissues and storing reserves, mainly as starch (Carbone et al., 2013; Kimak and Leuenberger, 2015). Âż Carbone, M.S., Czimczik, C.I., Keenan, T.F., Murakami, P.F., Pederson, N., Schaberg, P.G., Xu, X., Richardson, A.D., 2013. Age, allocation and availability of nonstructural carbon in mature red maple trees. New Phytol. 200, 1145–1155. https://doi.org/10.1111/nph.12448.

---

## Author Comment (AC5) · 25 May 2020

MMS & VD - Here we only include replies to comments that we have updated.

Referee 1 - General comments I also suggest the authors adding one section discussing uncertainty. Isotopic dendroclimatology is a subject based on chemical experiment. Unlike tree-ring width or density, the result of tree-ring d13C or d18O measurements are different to verified again from their core or disk samples, due to time consuming and great expense. It is possible to introduce mistakes during many steps of experiments, for example, impure cellulose and unreliable measurements caused by bad condition of the isotope ratio mass spectrometer. REPLY - Uncertainties do exist

for any kind of physical measurement, including tree ring width or density determination. We agree that in isotopic dendroclimatology, the chemical extraction of cellulose and the spectrometric measurements are critical steps. Impure cellulose and unreliable measurements yield bad data, which indeed more than likely diverge from climate. We will introduce a sentence of caution at the beginning of section 2.2 and refer to several papers devoted to good analytical practices. The text will read as follows: Âń A preliminary word of caution on tree-ring isotopic series is that the chemical extraction of cellulose and the spectrometric measurements are critical steps. Impure cellulose and unreliable measurements may yield erroneous data, which more than likely will diverge from climate. It is understood here that dendroisotopists should make sure to follow good analytical practices (see for instance Loader et al., 1997; Boettger et al., 2007; Wieloch et al., 2011; Kagawa et al., 2015; Andre-Hayles et al., 2019) but we do not think this subject has to be extensively discussed in this paper. Special comments are as follow.

Line 32, need a reference REPLY -We will add D'Arrigo et al., 2008. The text will read as follows: When correlations between climatic parameters and tree-ring proxies show periods of instability such that correlations weaken, become non-significant or change in signs, the relationship between proxies and climatic data shows a 'divergence' (D'Arrigo et al., 2008).

Lines 399-412 – multi-proxy approach and more climate noise? REPLY - We do not understand what the reviewer means. We write the contrary in the text: 'Indeed, combining proxies with the same dominant control, but different secondary controls, tends to accentuate the common climate signal'.

Table 1, check which one use pooling method. REPLY – good point. We identified the studies using pooled series (asterisks in Table 1 below), without identifying any specific common factors. Note that many of these studies validated that the use of pooled trees gave similar results to treatment of individual trees then merging mathematically. We will modify the end of section 3.1 : Note that when the pooling approach is envisaged for producing series of a specific tree species in a given region, verifying its reliability by comparison with averaged individual series is required prior to embracing the approach. This validation appears to allow produce isotopic series devoid of methodological artefacts (Table 1).

---

## Author Comment (AC6) · 25 May 2020

MMS & VD - Here we only include replies to referee 2 comments that we have updated.

Anonymous Referee #2

Line 125. The Model "MAIDEN" is not well explained, so I recommend explaining it a little bit, so the reader can understand what the model it's all about.

REPLY – Lines 121-125 define the general approach to mechanistic modeling, which applies to MAIDEN as well as to the other models of the kind. We do not want to place too much emphasis on MAIDEN, but following the suggestion of referee 2, we explain

briefly the main structure of MAIDENiso as follows (starting lines 129): Most models make forward predictions and allow verifying that the measured tree-ring isotopic trends compare well with the isotopic outputs modelled with the meteorological and non-meteorological inputs, and identifying processes behind isotopic responses. For instance, MAIDENiso is an expanded growth model which includes C and O modules. The model allows reproducing fractionation of carbon isotopes due to atmospheric $CO_2$ diffusion to the site of carboxylation, enzymatic photosynthesis and respiration, and estimates oxygen isotopes in precipitation, soil water and xylem water, and the fractionation in leaves due to evapotranspiration and biochemical formation of cellulose (details in Danis et al., 2012; Boucher et al., 2014).

Line 131 The citation for the Vaganov model it should be correctly cited, or add the papers where Vaganov published originally, then, of course, you can use other citations as usage examples.

REPLY – Good point. We will correct the name of the model and will refer to Vaganov et al., 2011. The text will read as follows: Âń Also considering process-based approaches, climatologists refer to the so-called proxy-system models (e.g., the Vaganov-Shashkin or VS model; Vaganov et al., 2011; Sánchez-Salguero et al., 2017).

Line 89 and Line 359 The percentage of oxygen isotope exchange during cellulose synthesis, as you mention, can indeed be variable. Recently there is a published paper addressing this same possibility and highlights some of the possible hypotheses that can be involved in such phenomena. Probably this is a reference you might be interested in exploring. New Phytologist (2020) doi: 10.1111/nph.16484

REPLY – Good point; this newly published reference is pertinent. We will add a citation to this article at former line 359 and add the full reference to the final list. The text will read: Âń However, this proportion may vary over growing seasons and longer periods due to relative humidity conditions (Gessler et al., 2009; Szejner et al., 2020). Âż

Line 253 The PIN correction of the pCO2 influence on the D13C discrimination should be double-checked. I think Gagen et al. 2007 made the first mention of the Pin correction that I know of. The Holocene, 17(4), 435–446. https://doi.org/10.1177/0959683607077012

REPLY – Correct. We add a citation to Gagen et al., 2007 before the citation to McCarroll et al., 2009, which describes six steps in the application of the method. The modified text will be: A widespread corrective approach uses a conditional, pre-industrial (pin) correction (Gagen et al., 2007). This six-steps non linear detrending of the low-frequency changes (McCarroll et al., 2009) better works when the measured 13C series starts before or at the beginning of the industrial period (1850), otherwise. . .

Line 206 Another recent publication Citation that you might be interested in exploring about age effects in Tree ring isotopes is form Xu et al. 2020. I think this is relevant to your review as it addresses the age-related effect concerning Climate reconstructions. 2020 Journal of Geophysical Research: Biogeosciences, 0–2. https://doi.org/10.1029/2019JG005513

REPLY – Good point; this freshly appeared reference is pertinent. We will add the text below with the citation to this article at the end of the last paragraph in section 3.2, and add the full reference to the final list. Text: Finally, in some cases, though there is no trend in the tree-ring isotopic series, the response to climate in the isotopic chronologies may be age-dependent. For instance, in Picea Schrenkiana from northwestern China, 18O and 13C values in trees under 125 years have stronger response to relative humidity than trees older than 270 years (Xu et al., 2020). The diminution of the strength of the correlations with tree age advocates for the incorporation of young trees only to develop a non-divergent composite chronology.

---

## Author Response (AR1)

**CP-2020-28 – Reply to comments from referees 1 and 2**

*Our replies are in blue italic fonts.*

**Referee 1 - General comments**

The manuscript reviews recent development of isotopic dendroclimatology, addressing possible divergence problem in tree-ring d13C and d18O. In my opinion, this kind work is very important when isotopic dendroclimatology has been paid more attention and plays more important role in high-resolution paleoclimate reconstruction. However, the current manuscript should be reorganized and concentrated in d13C. Physiological mechanisms between tree-ring d13C and d18O are quite different, therefore, comparing with tree-ring d13C, tree-ring d18O did not show recognizable effects from rising pCO2 (Lines 240-241) and pollution (Lines 424-425). Just as the authors said in Lines 514-515, tree-ring d18O is a more appropriate proxy for climate reconstruction. And, the phenomenon of divergence between the d18O and climate is really few worldwide.

**REPLY -** *Regarding presenting the review on $\delta^{18}O$ series, indeed, rising pCO$_2$ does not create divergence with climate for the $\delta^{18}O$ series, as we pertinently explain, but other causes do: changes in climatic regimes and pollution (sections 4.1 and 4.3). That is why we judge pertinent to keep the $\delta^{18}O$ series in this review (see Table 1).*

I also suggest the authors adding one section discussing uncertainty. Isotopic dendroclimatology is a subject based on chemical experiment. Unlike tree-ring width or density, the result of tree-ring d13C or d18O measurements are different to verified again from their core or disk samples, due to time consuming and great expense. It is possible to introduce mistakes during many steps of experiments, for example, impure cellulose and unreliable measurements caused by bad condition of the isotope ratio mass spectrometer.

**REPLY -** *Uncertainties do exist for any kind of physical measurement, including tree ring width or density determination. We agree that in isotopic dendroclimatology, the chemical extraction of cellulose and the spectrometric measurements are critical steps. Impure cellulose and unreliable measurements yield bad data, which indeed more than likely diverge from climate.*

*We have added a sentence of caution at the beginning of section 2.2 and refer to several papers devoted to good analytical practices, including the five new references cited. However, we do not think this subject has to be extensively discussed in this paper. The text on new lines 97-101 now reads as follows:* « A preliminary word of caution on tree-ring isotopic series is that the chemical extraction of cellulose and the spectrometric measurements are critical steps. Impure cellulose

and unreliable measurements may yield erroneous data, which more than likely will diverge from climate. It is understood here that dendroisotopists should make sure to follow good analytical practices (see for instance Loader et al., 1997; Boettger et al., 2007; Wieloch et al., 2011; Kagawa et al., 2015; Andre-Hayles et al., 2019).»

Some other uncertainties also exist. First is sampling strategy. We should understand what kind of tree could be used for climate reconstruction. As recommended by classical "The principle of limiting factors" (Fritts 1976), site selection is very important when one would employ trees to infer climate change. It is also important to isotopic dendroclimatology. Because mixed (deep phreatic water, shallow ground water, precipitation…) ground water may disturb tree-ring d18O (A tree in flowing figure), tree-ring d18O of B tree only absorb precipitation. Although cellulose d13C and d18O could be measured for any tree from any site, but for purpose of climate correlation, it should be carefully selected.

*REPLY – Thank you for raising that up. Indeed site selection is a crucial step in paleoclimate research. If trees and/or sites are not well selected, one of the main risks is that their $\delta^{18}O$ and/or $\delta^{13}C$ isotopic series show no significant relation with climate. In that case, reconstruction is not possible. Therefore, we judge that using poor criteria for site selection does not have to be dealt with in this paper as we assume that the readership is well aware of the initial step of adequate site selection.*

Second uncertainty may be introduced by different samples for measurement (extractive-free samples, α-cellulose, whole wood or holocellulose). And, for different chemical extraction methods (Green's method, Brendel's method…). Third uncertainty may be introduced by "pooling" or "not pooling".

*REPLY - Different extraction methods, as well as measurements produced in different laboratories with different spectrometers and procedures, should produce comparable results. There are very few inter-lab calibration experiments. To our knowledge the only study dealing with such a comparison was produced by the ISONET group (Boettger et al., 2007). Another one is in progress in the frame of the THEMES project conducted by one of the two authors (Daux, Andreu-Hayles et al., in progress). These inter-laboratory comparisons show that isotopic shifts may exist between laboratories (high correlations but different absolute value) due to differences in extraction methods, reference materials, instruments, etc. However, as long as the data included in an isotopic chronology have all been produced following the same protocols, by the same experimentalists, at the same laboratory, the data are consistent with one another and if they diverge from climate variations, the cause should be sought elsewhere.*

Special comments are as follow.

Lines 13-19, changes on physiology ($f_o$ and *peclet effect*...) should be mentioned here.
**REPLY** *-These items are covered in section 2.1. Here we cite the main CAUSES for divergences, not the mechanisms through which they operate. No change to these lines.*

Line 32, need a reference
**REPLY -***We have added D'Arrigo et al., 2008. The text now reads as follows:* « ... climatic data shows a 'divergence' (D'Arrigo et al., 2008). »

Lines 38-32, it is no need to descript growth divergence
**REPLY** *-The referee probably means lines 38-42. Here we just present some background using growth divergence. No change to these lines.*

Line 72, "concentration" is better than "pressure" , also in Line 236
**REPLY** *– Atmospheric $CO_2$ pressure or $pCO_2$ are well accepted and widely used. No change to these lines.*

Lines 94-95, need references
**REPLY** *-The references covering this topic are through equations 1-3, for which the citations are in the previous text. No change to these lines.*

Line 157, 5 year is not enough
**REPLY** *– Understood. The parenthesis underlines the fact that some researchers may opt for longer overlaps. No change to this line.*

Lines 175-177, one advantage for tree-ring isotope chronology is no need to detrending. If detrending for the isotope chronology, some climate signals may be lost.
**REPLY** *– We agree. That is exactly what the text explains. No change to these lines.*

Line 263, "1850s" is easy to understanding than "last 170 years"
**REPLY** *– We have modified the sentence (see new line 279), which now reads «... of rising $pCO_2$ since 1850.»*

Line 302, relative humidity and RH, repeat
**REPLY** *– we just present the abbreviation (RH) for relative humidity here. No change to this line.*

Line 376, tropic/
**REPLY** *– Tropical is a well accepted English adjective. No change to this line.*

Lines 399-412 – multi-proxy approach and more climate noise?

*REPLY - We do not understand what the reviewer means. We write the contrary in the text: 'Indeed, combining proxies with the same dominant control, but different secondary controls, tends to accentuate the common climate signal'. No change to this line.*

Section 3.3. there is only one sentence to state the situation of cellulose d18O (Line 240). I recommend to delete d18O discussion in this manuscript. In addition, removing effect of increasing CO2 from the d13C series has been discussed in many literatures. Please shorten this section.

*REPLY – It is true that the literature raised the issue abundantly, but there is still no consensus on how to approach and correct the problem. This manuscript designed to be a review article should cover the matter and section 3.3 intends to do just that. Concerning the effects on $\delta^{18}O$ values, it is worth explaining which articles address the potential $pCO_2$ effects, even if nil or minimal. In addition, referee 2 pertinently suggests to integrate new references to this section. So we decide not to shorten the section.*

Table 1, check which one use pooling method.

*REPLY – good point. We identified the studies using pooled series (asterisks in Table R below), without identifying any specific common factors. Note that many of these studies validated that the use of pooled trees gave similar results to individual trees merged mathematically. We have kept Table 1 in its original form, but modified the end of section 3.1 (new lines 196-199): «*Note that when the pooling approach is envisaged for producing series of a specific tree species in a given region, verifying its reliability by comparison with averaged individual series is required prior to embracing the approach. This validation appears to allow producing isotopic series devoid of methodological artefacts (Table 1).»

**Table R. Reported critical divergences of correlations between isotopic results and instrumental climatic series (other than sampling, stand dynamics and juvenile effects).**

| Isotopes | Climate Parameters | Tree species | Causes | Region | Author(s) |
|---|---|---|---|---|---|
| $\delta^{13}C$ | Summer T | *Quercus robur* | CC: longer growth season | Eastern England | Aykroyd et al., 2001 |
| $\delta^{13}C$, $\delta^{18}O*$ | Summer T, Pc | *Quercus petraea; Pinus sylvestris* | CC: physiological adaptation to higher T, change in moisture origin | Switzerland | Reynolds et al., 2007 |
| $\delta^{13}C*$ | Summer T | *Pinus sylvestris* | CC: earlier Summer | Eastern Finland | Hilasvuori et al., 2009 |
| $\delta^{13}C*$ | Summer T& Pc | *Quercus robur, Pinus sylvestris* | Poll: $SO_2$ from close emitter | Southeastern England | Rinne et al., 2010 |
| $\delta^{13}C$ | Summer cloud cover, T | *Pinus sylvestris* | CC: AO, decoupling of T and radiations | Northwestern Norway | Young et al., 2010 |
| $\delta^{13}C$, $\delta^{18}O^p$ | Tmax, RH | *Larix decidua* | CC: drier climate; deeper soil water | French Alps | Daux et al., 2011 |
| $\delta^{13}C$, $\delta^{18}O*$ | Summer T & Pc | *Pinus sylvestris* | CC: change in T, irradiance & cloud circul. | N. boreal zone | Seftigen et al., 2011 |
| $\delta^{13}C*$ | Summer T& Pc | *Larix decidua* | Poll: traffic/vehicles | Italian Alps | Leonelli et al., 2012 |
| $\delta^{18}O$ | Summer Pc | Pinus halepensis | CC: increase of drought; deeper soil water | Greece | Sarris et al., 2013 |
| $\delta^{13}C$ | *No link* | *Juniperus virgianiana* | Poll: distant $SO_2$ emitters | Appalachians, USA | Thomas et al., 2013 |
| $\delta^2H$, $\delta^{13}C$, $\delta^{18}O$ | RH | *Abies alba* | Poll: distant $SO_2$ emitters | Southwestern Germany | Boettger et al., 2014 |
| $\delta^{13}C*$ | RH, T | *Abies georgei* | CC: water stress | Western China | Liu et al., 2014 |
| $\delta^{13}C*$ | Tmax | *Picea mariana & glauca* | Poll: oil sands mining operations | Alberta, Canada | Savard et al., 2014 |

| $\delta^{13}C, \delta^{18}O$ | Summer T
Spring Pc | *Picea mariana* | CC; NAO
longer growth season | Northeastern
Canada | Naulier et al., 2015b |
|---|---|---|---|---|---|
| $\delta^{13}C*$ | Spring-Sum. T | *Sabina przewalskii* | CC: change in cloud circulation | Tibet | Wang et al., 2016; 2019 |
| $\delta^{13}C, \delta^{18}O$ | VPD | *Pinus ponderosa* | CC: increase of drought | Southw. USA | Szejner et al., 2018 |
| $\delta^{18}O$ | Spring AO,
spring NAO | *Cryptomeria japonica* | CC: spring AO-EASM changes | Northeastern
Japan | Sakashita et al., 2018 |
| $\delta^{13}C$ | *No link* | *Picea rubens* | Poll: distant $SO_2$ emitters | Appalachians,
USA | Mathias & Thomas, 2018 |
| $\delta^{18}O*$ | May-July T, RH,
PDSI | *Abies forrestii* | CC: change in moisture origin | Southwestern
China | An et al., 2019 |
| $\delta^{13}C$ (WUE)$^P$ | Summer Tmax | *Picea mariana & glauca* | Poll: bitumen mining, metal smelter emissions, global $CO_2$ rise | Alberta &
Québec, Canada | Savard et al., 2020 |

T : temperature. Tmax : maximum temperature. RH : relative humidity. Pc: precipitation. VPD: vapour pressure deficit. AO : Arctic oscillations. NAO : North Atlantic Oscillations. PDSI: Palmer drought severity index. CC : climate change. Poll : pollution stress. EASM: East Asian summer monsoon. * indicates series exclusively involving pooled tree rings; $^P$, series partially composed of pooled rings.

**Anonymous Referee #2**

**Comment on the review from referee 1**

First, I would like to thank the authors for conducting this synthesis. The authors did a great job on synthesizing and explaining all the different sources of divergence caused by multiple factors recorded in Carbon and Oxygen isotopes in the wood.

At the beginning of this review, the authors explain the differences between the "divergence" topic in the tree ring community versus the divergence that can be found in isotopic measurements in Tree rings. I see why the Authors are attributing the term divergence to the examples they show. However, it is not clear if the term divergence is the correct term. It is fine to use this term as long the authors make sure that they are referring to the divergence to the climate signal and eventually highlight that this "issue" falls into the problems we as scientists have when we want to interpret the isotopic records in Tree rings. I do appreciate the sections where they make recommendations and a strong call to the good practices so future researchers can take this advice to minimize the chances of losing the climatic signal.

*REPLY – We sincerely thank referee 1 for the constructive comments and suggestions compiled above and below. Regarding the usage of 'divergence', we agree with the referee that this term should be restricted to describing tree-ring isotopic departures from climatic parameters. That is what we rigorously do in the manuscript. The introduction explains lines 30-32:* «When correlations between climatic parameters and tree-ring proxies show periods of instability such that correlations weaken, become non-significant or change in signs, the relationship between proxies and climatic data shows a 'divergence'. » *Further down (lines 44-45):* «The present article deals with the 'isotopic divergence', which we define here as the middle- to long-term (>10 years) loss or change in signs of correlations between a climatic parameter and tree-ring isotopic ratios ($\delta^{13}C$, $\delta^{18}O$, or rarely $\delta^2H$). » *So no changes needed.*

**Anonymous Referee #2**
**Some other suggestions**

One comment I should mention is that the review is highly focused on climate reconstructions, while the ecophysiological responses to environmental cues are somewhat left a little bit on the side, as something that is dampening or disrupting the climate signal.

*REPLY – It is right to reckon that the article focuses on the tree-ring isotopes-climate relationships with the main purpose of climatic reconstruction as explained in the introduction (lines 61-63):* «Given the need for careful assessments of isotopes as climate proxies for various regional contexts and tree species, this synthesis of the up-to-date information on isotopic divergences aims at: (1) describing the main isotopic divergence types and discussing their potential causes, and (2) reviewing research avenues to identify them and account for them (Table 2). »

*On one hand, wide ecological changes are not included in the manuscript on purpose as we want to restrict the review to TR isotopic divergences due to direct tree responses. On the other hand, we refer to the ecophysiological approaches for assessing tree responses to environmental changes as part of eventual solutions for circumventing some isotopic divergence issues (see for instance Section 5; paragraph before last). Therefore, we did not take actions in response to this comment.*

So, the more specific comments are more targeted to references in the literature (given this manuscript is a review) plus some other clarifications if the author agrees.

Line 125. The Model "MAIDEN" is not well explained, so I recommend explaining it a little bit, so the reader can understand what the model it's all about.

*REPLY – Lines 124-129 define the general approach to mechanistic modeling, which applies to MAIDEN as well as to the other models of the kind. We do not want to place too much emphasis on MAIDEN, but following the suggestion of referee 2, we now explain briefly the main structure of MAIDENiso as follows (new lines 129-135): «* Most models make forward predictions and allow verifying that the measured tree-ring isotopic trends compare well with the isotopic outputs modelled with the meteorological and non-meteorological inputs, and identifying processes behind isotopic responses. For instance, MAIDENiso is an expanded growth model which includes C and O modules. The model allows reproducing fractionation of carbon isotopes due to atmospheric $CO_2$ diffusion to the site of carboxylation, enzymatic photosynthesis and respiration, and estimates oxygen isotopes in precipitation, soil water and xylem water, and the fractionation in leaves due to evapotranspiration and biochemical formation of cellulose (details in Danis et al., 2012; Boucher et al., 2014; Lavergne et al., 2017*). »*

Line 131 The citation for the Vaganov model it should be correctly cited, or add the papers where Vaganov published originally, then, of course, you can use other citations as usage examples.

*REPLY – Good point. We have corrected the name of the model and now refer to Vaganov et al., 2011. The text now reads as follows (new lines 140-141): «* … refer to the so-called proxy-system models (e.g., the Vaganov-Shashkin or VS model; Vaganov et al., 2011; Sánchez-Salguero et al., 2017).»

Line 89 and Line 359 The percentage of oxygen isotope exchange during cellulose synthesis, as you mention, can indeed be variable. Recently there is a published paper addressing this same possibility and highlights some of the possible hypotheses that can be involved in such phenomena. Probably this is a reference you might be interested in exploring. New Phytologist (2020) doi: 10.1111/nph.16484

*REPLY – Good point; this newly published reference is pertinent. We have added a citation to this article at new line 377 and add the full reference to the final list. The text now reads: «* However, this proportion may vary over growing seasons and longer periods due to relative humidity conditions (Gessler et al., 2009; Szejner et al., 2020). »

Line 253 The PIN correction of the pCO2 influence on the D13C discrimination should be double-checked. I think Gagen et al. 2007 made the first mention of the Pin correction that I know of. The Holocene, 17(4), 435–446. https://doi.org/10.1177/0959683607077012

*REPLY – Correct. We have added a citation to Gagen et al. (2007) before the citation to McCarroll et al. (2009), which describes six steps in the application of the method. The modified text on new lines 268-270 is: «* A widespread corrective approach uses a conditional, pre-industrial (pin) correction (Gagen et al., 2007). This six-steps non linear detrending of the low-frequency changes (McCarroll et al., 2009) …».

Line 206 Another recent publication Citation that you might be interested in exploring about age effects in Tree ring isotopes is form Xu et al. 2020. I think this is relevant to your review as it addresses the age-related effect concerning Climate reconstructions. 2020 Journal of Geophysical Research: Biogeosciences, 0–2. https://doi.org/10.1029/2019JG005513

*REPLY – Good point; this freshly appeared reference is pertinent. We have added the text below with a citation to this article at the end of the paragraph before last in section 3.2 (new lines 238-242), and added the full reference to the final list.*
« Finally, in some cases, though there is no trend in the tree-ring isotopic series, the response to climate in the isotopic chronologies may be age-dependent. For instance, in *Picea Schrenkiana* from northwestern China, $\delta^{18}O$ and $\delta^{13}C$ values in trees under 125 years have a stronger response to relative humidity than trees older than 270 years (Xu et al., 2020). A diminishing strength of the correlations with tree age advocates for the incorporation of young trees only to develop a non-divergent composite chronology. »

Xu, G., Wu, G., Liu, X., Chen, T., Wang, B., Hudson, A., 2020. Age-related climate response of tree-ring $\delta$ 13 C and $\delta$ 18 O from spruce in northwestern China, with implications for relative humidity reconstructions 0–2. https://doi.org/10.1029/2019JG005513

Lien 262. I agree that there is no overarching consensus over how to correct the pCO2 effects on the discrimination of 13C. But I find a bit troubling this sentence "A wise approach is to test the various corrective methods and assess the performance of the resulting series with climatic reconstruction model." This statement is for me, suggesting that we should select the best fit to climate. I think this is a bit biased and undermined the fact that we still do not fully understand how the pCO2 is affecting gs and A. so I think this part needs to be careful on not incentivize researchers to select the best fit, but instead, incentive to investigate what is the mechanisms and how the pCO2 is or not affecting the Carbon chronologies. Then I suggest reviewing Global Change Biology, 22(2), 889–902. https://doi.org/10.1111/gcb.13102

*REPLY – We agree. The text is modified as follows on new lines 276-279: «* A wise approach is to investigate the potential influence of $pCO_2$ on isotopic ring series and the gas-exchange response mechanisms in trees prior to selecting a corrective method (Voelker et al., 2016; Savard et al., 2020). »

Line 287 I think this part needs this reference. Dorado-Liñán, I et al. 2016. Climate Dynamics, 47(3–4), 937–950. https://doi.org/10.1007/s00382-015-2881-x

*REPLY – We agree. We have added the reference and the text modified on new lines 306-309 now reads: «* … (Young et al., 2010), the Northern boreal zone (Seftigen et al., 2011) and Northern Spain (Dorado-Liñan et al., 2016) depicted divergences between temperature records and $\delta^{13}$C series of pines (*Pinus sylvestris* or *Pinus uncinata*) during episodes of decoupling between irradiance and temperature linked to either changes in large scale atmospheric circulation (in the first two references), or large volcanic eruptions (in the third one). »

Line 345 This reference also can be useful here Carbone, M. S. et al. 2013, The New Phytologist, 200(4), 1145–55. https://doi.org/10.1111/nph.12448

*REPLY – We agree. We have added the reference and the text modified on new line 361 now reads: «*… mainly as starch (Carbone et al., 2013; Kimak and Leuenberger… »